# The role of water mobility on water-responsive actuation of silk

Darjan Podbevšek [1,2], Yeojin Jung[1,2], Maheen K. Khan[1,2], Honghui Yu [3], Raymond S. Tu [1,2] ✉ & Xi Chen [1,2,4] ✉

Biological water-responsive materials that deform with changes in relative humidity have recently demonstrated record-high actuation energy densities, showing promise as high-performance actuators for various engineering applications. However, there is a lack of theories capable of explaining or predicting the stress generated during water-responsiveness. Here, we show that the nanoscale confinement of water dominates the macroscopic dehydration-induced stress of the regenerated silk fibroin. We modified silk fibroin's secondary structure, which leads to various distributions of bulk-like mobile and tightly bound water populations. Interestingly, despite these structure variations, all silk samples start to exert force when the bound-to-mobile (B/M) ratio of confined water reaches the same level. This critical B/M water ratio suggests a common threshold above which the chemical potential of water instigates the actuation. Our findings serve as guidelines for predicting and engineering silk's WR behavior and suggest the potential of describing the WR behavior of biopolymers through confined water.

Water-responsive (WR) materials are a class of solid matter capable of deforming in response to changes in the relative humidity (RH) by absorbing or releasing water[1,2]. While widely present in nature[2], these materials have gained increased interest in recent years, as they show great promise in developing high-performance actuators for robotics[3,4], smart textiles[5], shape-adaptive electrodes for bioelectronic interfaces[6], autonomous structures[7], passive regulators[8], and energy conversion devices[3,9–11]. Certain nanoscale biological and bio-derived materials, such as bacterial spores[9,12] and peptidoglycan[3], possess exceptionally high WR energy densities of up to 72.6 MJ/m³, orders of magnitudes higher energy densities than existing actuator materials and natural muscles[1,3]. Another notable bioderived WR material is regenerated silk fibroin (RSF) from widely available silkworm (*Bombyx mori*) cocoons[13–15]. A modified version of RSF recently demonstrated WR energy density of 3.1 MJ/m³ [15]. While this value is approximately an order of magnitude lower than the highest reported value of peptidoglycan, RSF offers advantageous applicability due to

its versatility in processing, scalability and ability to form films, scaffolds, particles, and fibers using well-established methods[16]. Despite the impressive WR performance of these biomaterials, a consensus on the fundamental physics governing their WR properties has not yet been reached[2]. Additionally, there is a lack of reliable methods for predicting their swelling/shrinking strain and stress when subjected to RH changes, which impedes the use and engineering of WR materials[2].

Over the past two decades, considerable efforts have been directed towards the quantitative prediction of a material's water-responsiveness. Much of this effort frames the adsorption-induced deformation of a material in the context of macroscopic thermodynamics[17], where the mechanical deformation is driven by the difference in chemical potential between water confined within the material and water vapor in the surrounding environment[17,18]. While these attempts can describe swelling behaviors for some porous materials with relatively low WR strain[7,17], it remains difficult to predict the WR behaviors of biomaterials and other non-porous soft matter,

[1]Advanced Science Research Center (ASRC) at the Graduate Center, The City University of New York, 85 St. Nicholas Terrace, New York, NY 10031, USA. [2]Department of Chemical Engineering, The City College of New York, 275 Convent Ave, New York, NY 10031, USA. [3]Department of Mechanical Engineering, The City College of New York, 275 Convent Ave, New York, NY 10031, USA. [4]PhD Programs in Chemistry and Physics, The Graduate Center of the City University of New York, 365 5th Ave., New York, NY 10016, USA. ✉e-mail: tu@ccny.cuny.edu; xchen@gc.cuny.edu

particularly those with strong material-water interactions and bonding reorganization that lead to a dramatic change in the chemical potential of confined liquids during WR actuation[2,17]. At the molecular scale, most protein-based WR materials are surrounded by a hydration shell[19,20], where the water forms an ordered hydrogen-bonding (H-bonding) network. This network is often described as an intermediate phase between disordered bulk-like and structured ice-like water[21,22]. While definitions vary, these confined water molecules are often referred to as "bound" water, meaning water molecules that are within short distances and interact strongly with the material's surface[19,20], as opposed to "mobile" water, that has little or no interaction with the material[21]. Studies have shown that the amounts of bound and mobile water are always relevant[23]. In the hydration shell, bound water slowly evolves and forms H-bonds with biomolecules and other water molecules, while mobile water forms weaker H-bonds and lacks interactions with biomolecules[24]. These hydration shells of bound and mobile water were found to be critical for structural stability[24,25] and bio-functionality[19], such as enzyme functionality[20], anti-freeze protein activity[26] and structural reconfigurations[27–31]. Specifically, bound water plays a crucial role in achieving a consistent thermodynamic understanding of the instability of structured proteins[30], transferable protein design[29], and free-energy barriers toward protein aggregation[31]. Prior studies on WR materials, including bacterial spores[32], peptidoglycan[3], RSF[15], and peptide-based crystals[33,34], show evidence that high-performance WR actuation is related to the properties of water in these hydration shells. Examples of these include dehydration-enhanced H-bonding networks that promote the forceful WR actuation of peptide crystals[33] and the hydration forces resulting from water confinement in nanoscale pores that dominate the nonlinear mechanical and WR properties of spores[32]. These observations prompted us to explore WR actuation through the hydration structures of confined water in WR biopolymers.

In this study, we use cast RSF films as a model material to study how the water structure within biopolymer networks influences their dehydration-induced water-responsiveness (Fig. 1a). The properties of confined water were modified by altering the secondary structure of the RSF films through post-treatment methods[13]. The resulting RSF films (Fig. 1b), designated Silk-H, Silk-M and Silk-L, exhibit decreasing amounts of water adsorption: 70.1%, 45.5% and 19.1%, respectively, when exposed to RH changes from 10% to 90%. The adsorbed water in these three types of RSF films also exhibit distinct differences in the bound-to-mobile (B/M) water ratios during hydration and dehydration at different RH (Fig. 1c). Despite the similar WR strain shared by all three kinds of RSF films in response to RH changes (Fig. 1d), RSF silk films demonstrate markedly different WR stress characteristics, with force exertion starting at different RH levels (46% RH for Silk-H, 66% RH for Silk-M, and 77% RH for Silk-L) (Fig. 1e). Using the Zener model to analyze the WR behaviors, we observed that all the RSF films undergo a transition from liquid-like to solid-like states, and the force exertion starts at this phase transition point during dehydration (Fig. 1f–h). More interestingly, such viscous-to-rubbery transitions in all RSF films occur at a similar B/M water ratio between 0.8 and 0.9 (0.84, 0.80 and 0.88, for Silk-H, -M and -L, respectively), beyond which the stress increase to an asymptotic plateau (Fig. 1i). This critical B/M water ratio suggests a common threshold, where the stress induced by the chemical potential of water at the molecular level can be transferred to macroscopic stress of the RSF material regardless of secondary structures. Such findings show the dominant role that water structure plays in RSF's WR actuation, and the findings also provide guidelines for predicting or engineering the WR behaviors for a broad range of biopolymers through confined water.

## Results

The RSF biopolymer offers a unique level of structural flexibility, as the protein's secondary structure can be altered to control both the mechanical and WR properties[13–15,35]. For example, within the silk fibroin backbone, the secondary structure of the heavy chain gives rise to both soft, hydrophilic amorphous (intrinsically disordered) regions and stiff, hydrophobic β-sheets nanocrystalline regions[35,36]. The ratio between these regions can be manipulated through post-treatment

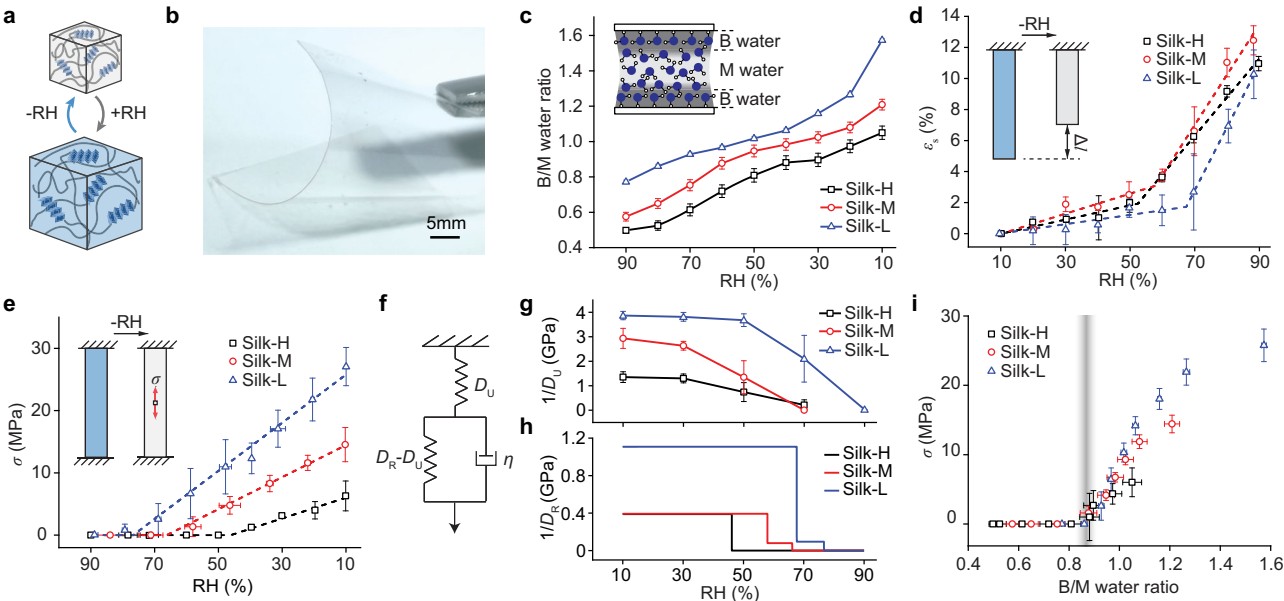

**Fig. 1 | Water-responsiveness of RSF films. a** WR materials swell and shrink in response to RH changes. **b** RSF films with tunable properties of confined water. **c** B/M water ratios of Silk-H, Silk-M, and Silk-L as a function of RH. Error bars represent the propagated SE of the deconvolution area fit. Dehydration-induced WR strain (**d**) and stress (**e**) of Silk-H, Silk-M, and Silk-L as a function of RH during dehydration. Data are presented as mean ± SD ($n = 4$ for (**d**) and $n = 3$ for (**e**)). The dash lines in (**d**) and (**e**) are the segmented linear fit of the data. **f–h** The Zener model used to analyze the WR viscoelastic behavior of RSF films. **g, h** show the changes in unrelaxed and relaxed moduli of $E_U$ (or $1/D_U$) and $E_R$ (or $1/D_R$), respectively, as RH decreases from 90% to 10%. Data in (**g**) are presented as mean ± SD ($n = 3$). **i** WR stress at their corresponding B/M water ratio plots of Silk-H, Silk-M, and Silk-L. Error bars represent the propagated SE and SD of the B/M water ratio and WR stress, respectively.

modifications[13,16]. To modify the secondary structures of RSF and the properties of nanoconfined water, we cast the RSF solution into thin films and, subsequently, employed water- or methanol-based treatments to increase the volume ratio of the crystalline domains within these films. We designated the three types of resulting samples as Silk-H, Silk-M and Silk-L, corresponding to the high (untreated), medium (water-treated) and low (methanol-treated) water adsorption capabilities, respectively.

## Dehydration-induced strain and stress of RSF

To evaluate the WR properties of these three types of RSF films, we first conducted WR strain tests by comparing the length of these films at different RH (normalized to 10% RH) without applying any external load. We noted that, as RH decreases from 90% to 10%, Silk-H, Silk-M, and Silk-L exhibited a similar WR free strain characteristic (Fig. 1d), each demonstrating a maximum strain of between 10.3% to 12.4%, regardless of post-treatments applied to the samples. However, when we evaluated their WR blocking stress by fixing the length of the sample during dehydration and measuring the evolution of stress, we found that these RSF samples exhibited significantly different characteristics. As the environmental RH decreases from 90% to 10%, the ultimate dehydration-induced stress of Silk-H, Silk-M, and Silk-L reaches 6.3 MPa, 14.5 MPa, and 27.1 MPa, respectively (Fig. 1e). This trend is consistent with the previously observed increase in WR energy density with the rising crystallinity of RSF[13]. We found that, during dehydration, these samples began to exert notable forces at different RH levels: 77% RH (Silk-L), 66% RH (Silk-M), and 46% RH (Silk-H) (Fig. 1e). We also measured the Young's modulus of RSF films at different RH. We observed that all RSF films exhibit viscoelastic behaviors, with the unrelaxed Young's moduli for Silk-H, Silk-M, and Silk-L decreasing from 1.35 GPa, 2.93 GPa, and 3.87 GPa, respectively, at 10%

RH to nearly zero at 90% RH. These viscoelastic behaviors are comparable to those reported in *Antheraea pernyi* moth silk fibers[37], which exhibit a plasticizing effect with increasing water content[35].

We modeled the observed WR and mechanical properties of RSF films using the Zener model, which is frequently used to describe the viscoelasticity of a material (Fig. 1f). In this model, $E_U$ (or $1/D_U$) is the short-time unrelaxed modulus, corresponding to the measured Young's modulus from the immediate response, while $E_R$ (or $1/D_R$) represents the relaxed modulus of the viscous components of the RSF films. By fitting the measured WR strain, stress, and Young's modulus to the Zener model, we found that both the unrelaxed and relaxed moduli show strong non-linear behaviors during dehydration (Fig. 1g, h). Notably, the relaxed modulus of $E_R$ shows a dramatic change from zero at 90% RH to 1.108 GPa, 0.393 GPa, and 0.389 GPa for Silk-L, Silk-M, and Silk-H, respectively (Fig. 1h). This change suggests the transition of RSF from a viscous state to a rubbery state, and this phase transition corresponds to where the RSF material initially exerts force. Both relaxed and unrelaxed moduli show weak dependence on the RH in the rubbery state, especially for Silk-L.

## Secondary structure-dependent water adsorption of RSF

To quantify the volume fractions of the secondary structures of RSF films, including β-sheet, random coil, α-helix, and β-turn, we deconvoluted the Amide I and II bands obtained through attenuated total reflection (ATR) Fourier transform infrared (FTIR) spectroscopy using a previously reported protocol[38] (Fig. 2a). We observed similar deconvoluted secondary structures from Amide I and Amide II band (Fig. S1), but only the Amide I is used for the secondary structure quantification[38] (Fig. 2b). We observed a significant decrease in the volume fraction of the random coil in the amorphous phase: 31.3% (Silk-H), 17.2% (Silk-M), and 7.5% (Silk-L). Simultaneously, there is an

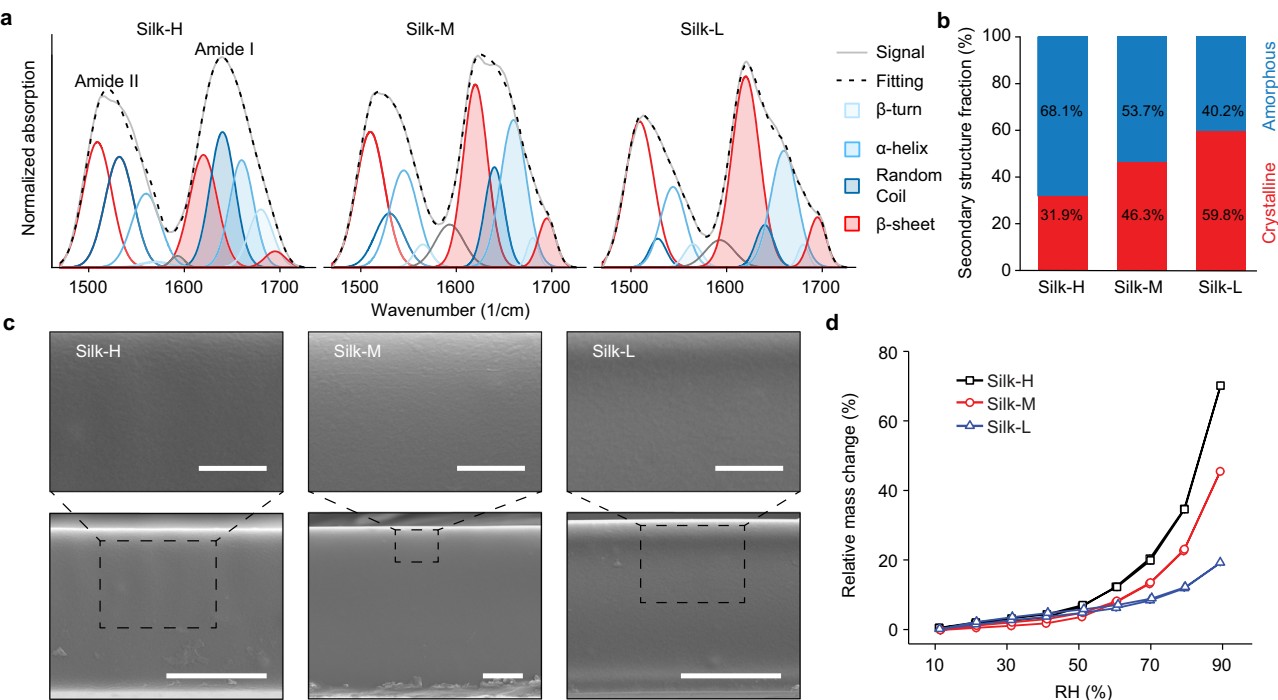

**Fig. 2 | Secondary structure, SEM, and water sorption isotherms of the RSF samples. a** Normalized ATR-FTIR spectra of Amide I (1650/cm peak) and II (1520/cm peak) bands and the deconvolution peaks for Silk-H, Silk-M, and Silk-L samples. The deconvolution peaks are centered at 1510/cm (β-sheet), 1530/cm (Random coil), 1540/cm (α-helix), 1565/cm (β-turn) for Amide II peak, 1592/cm (baseline correction—see "Methods" section), and 1620/cm (β-sheet—inter-/intra-molecular), 1640/cm (α-helix), 1655/cm (Random Coil), 1680 (β-turn), 1695/cm (β-sheet—

intermolecular) for Amide I peak. **b** Crystalline and amorphous fractions of Silk-H, Silk-M, and Silk-L were determined from the areas of the deconvolution peaks of the Amide I peak. **c** SEM images of cryo-fractured cross-sections of RSF films. The scale bars in the upper and lower images are 1 μm and 3 μm, respectively. **d** Relative mass change as a function of RH measured by DVS. Data are presented as mean ± SD (n = 3 hydration/dehydration cycles).

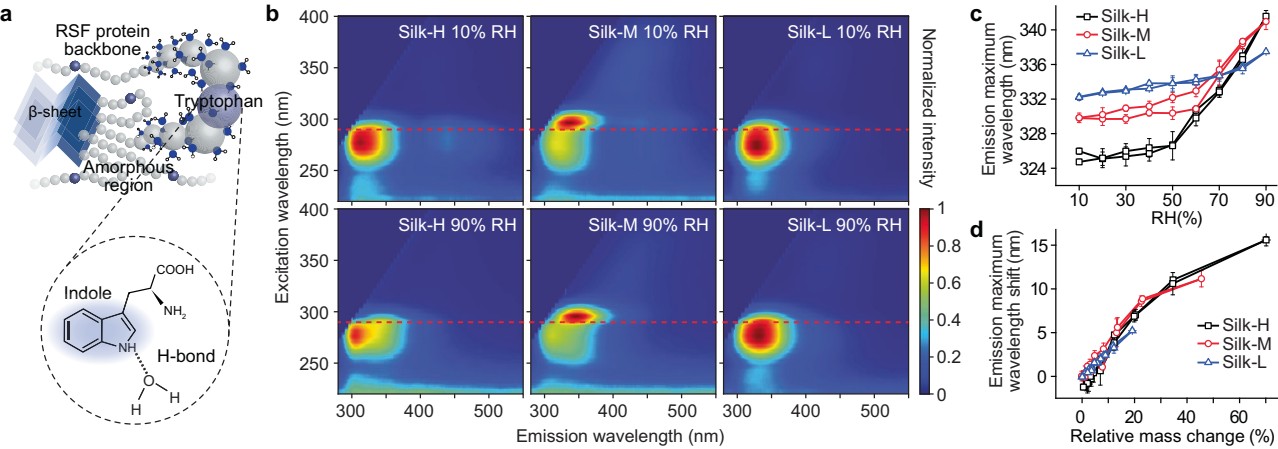

**Fig. 3 | Tryptophan fluorescence shift during RH cycle. a** Schematic depicting Tryptophan's location in the amorphous regions of RSF has fluorescence properties that are highly sensitive to the hydration state. **b** The emission/excitation map of fluorescence of RSF at 10% and 90% RH. The red dashed line indicates the 290 nm excitation wavelength. **c** The emission maximum of RSF samples during hydration and dehydration. **d** The emission maximum shift as a function of water adsorption, datasets normalized to the 10% RH point. Data in (**c**) and (**d**) are presented as mean ± SD (*n* = 3 hydration/dehydration cycles).

increase in β-sheet nano-crystallites, rising from 31.9% in Silk-H to 46.3% in Silk-M and reaching 59.8% in Silk-L.

The cross-section scanning electron microscope (SEM) images and atomic force microscopy (AFM) topographies show that all RSF films share similar non-porous solid structures (Figs. 2c and S12). The water sorption isotherms of these RSF samples were characterized using a dynamic vapor sorption (DVS) device (Figs. 2d and S3). As the RH was cycled between 10% and 90% in 10% increments, the ultimate water adsorptions of Silk-H, Silk-M and Silk-L were 70.1%, 45.5% and 19.1% of the initial mass measured at 10% RH, respectively (Fig. 2d). The obtained water sorption isotherms show low hysteresis, and they belong to a tradition type III isotherms[39,40], providing further evidence of the non-porous nature of RSF films. Thus, the difference in the water uptake of RSF should be attributed to their secondary structures. These measured water adsorptions are also consistent with those reported in literature for RSF with similar secondary structures[13], namely, decreasing with the increased β-sheet crystalline domain fraction. Given the hydrophobic nature of the crystalline regions, it is likely that the decreased amount of water adsorption in RSF results from a reduced volume fraction of the hydrophilic amorphous regions.

### RSF tryptophan fluorescence and molecular-scale hydration
The fluorescence of tryptophan residues further indicates that the adsorbed water is localized within the amorphous regions in RSF (Fig. 3). Silk fibroin possesses intrinsic fluorescence properties, stemming from the tyrosine and tryptophan residues[41]. Tryptophan fluorescence is highly sensitive to its environment and is often used for studying protein conformation via the fluorescence emission peak shift[42], while tyrosine emission has no such peak shift[41]. The benzene and pyrrole ring making up the indole part (chromophore) of the tryptophan residue, exhibits fluorescence properties that can red shift or blue shift depending on the electron density shift along the excited electronic state transition dipole[43] (Fig. 3a). The degree of hydration around the protein backbone is expected to be proportional to the red-shifting of tryptophan fluorescence, similar to the solvent exposure experiments in proteins solvation studies[43].

Tryptophan residues in *Bombyx mori* silk are found exclusively in the linkers (irregular phase) of the heavy chains (Fig. 3a)[36], and they can yield localized information of the hydration state in the amorphous regions of RSF samples. When RH changes between 10% and 90%, all RSF samples exhibit a shift in their tryptophan emission wavelength (Fig. S4). To quantify the tryptophan emission shift and analyze the

hydration state of RSF's amorphous regions, we compared the emission spectra of RSF using an excitation wavelength of 290 nm (red dashed line in Figs. 3b and S4). During hydration, the emission maximum of Silk-H, Silk-M and Silk-L shows a red shift from 325 nm, 329 nm and 332 nm at 10% RH to 341.5 nm, 341 nm and 337.5 nm at 90% RH, respectively (Fig. 3c). We noted that the emission maximum of RSF at 10% RH is higher than that of tryptophan in non-polar conditions (305 nm)[43], which can be attributed to the interactions between tryptophan and water confined in RSF even at low RH conditions. Compared to Silk-H and Silk-M, Silk-L has a relatively smaller emission shift, which could be associated with the saturation of the reduced amount of hydrophilic regions exposed to water, after the methanol treatment (Fig. 3c). When normalizing the emission maximum to the initial conditions at 10% RH in the hydration stage and relating the shift in emission maximum to the amount of water adsorption of RSF at different RH, we found that all RSF samples share a similar relationship between the emission shift and water uptake (Fig. 3d). Considering that tryptophan residues are exclusively located in the amorphous regions[44], this observed correlation between the emission shift and water adsorption across all RSF samples suggests that the adsorbed/desorbed water is mainly located in the amorphous regions. Otherwise, the tryptophan emission would show less shift per water molecule uptake in RSF with a higher degree of crystallinity.

### H-bonding network characterization by FTIR
To quantitatively understand the properties and the H-bonding networks of water confined in these three RSF samples under different RH conditions, we performed transmission FTIR experiments, which allowed us to estimate the fraction of tightly bound water and bulk-like mobile water from the O-H stretching band (Fig. 4). The broad O-H stretching band from 2500 cm⁻¹ to 3700 cm⁻¹, attributed to the water content within the RSF samples, is known to be highly sensitive to the state of the H-bond network[45]. This O-H stretching band can be cataloged into five populations of H-bonds, representing a convolution of five Gaussian peaks (3000 cm⁻¹, 3210 cm⁻¹, 3290 cm⁻¹, 3410 cm⁻¹, and 3550 cm⁻¹)[45–49] (Figs. 4a, b and S5–S7). These populations correspond to four different levels of acceptor (A) and donor (D) H-bonding configurations (AADD, AD, AAD, ADD)[45] as well as the water-solid interaction[50]. We estimate the fractions of these five populations of water in RSF samples at different RH by normalizing the areas of the deconvoluted peaks to the areas of each sample's O-H stretching band at 10% RH (Fig. 4c). As RH increases from 10% to 90%, the total amount

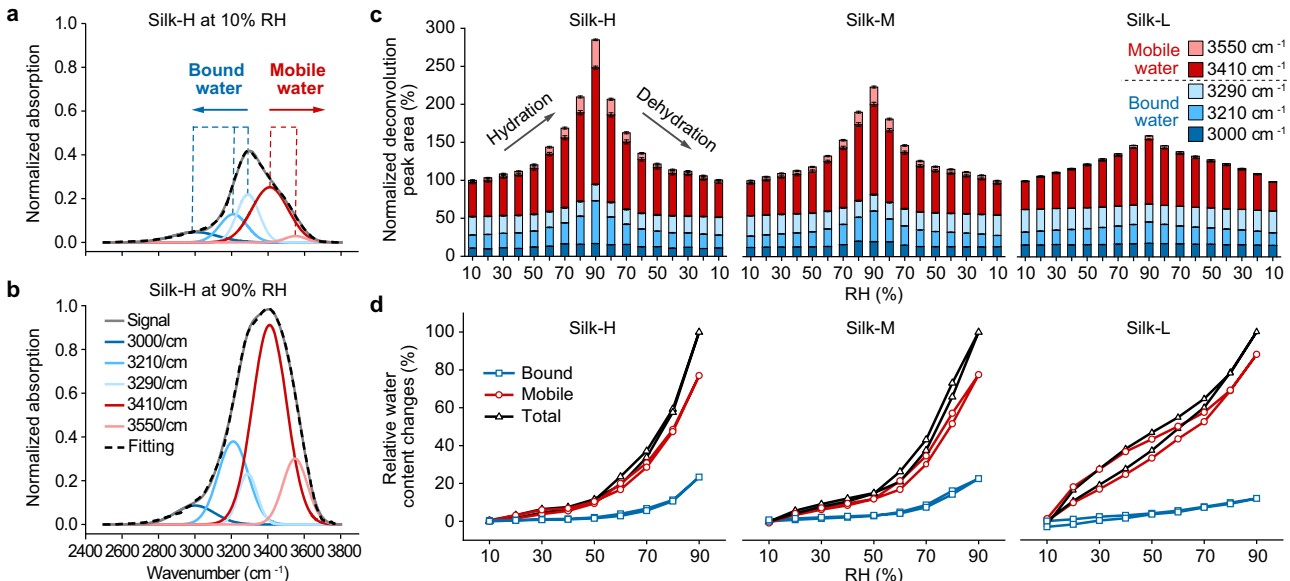

**Fig. 4 | Bound and mobile water contents in RSF.** The FTIR O-H stretching spectra of Silk-H at 10% RH (**a**) and 90% RH (**b**). **c** The relative contents of mobile and bound water in Silk-H, Silk-M, and Silk-L, normalized to the total water content of each RSF sample at 10% RH. Data are presented as the deconvolution peak area ± SE of the fit. **d** The relative content changes of total, mobile, and bound water normalized to the total water content of each RSF sample at 90% RH.

of the five water populations obtained from the FTIR deconvolution clearly shows an increase in the total O-H stretching areas of 184.9%, 123.2% and 59.3%, for Silk-H, Silk-M, and Silk-L, respectively. These values are proportional to the ultimate water adsorptions measured by DVS. Note that the DVS water sorption isotherms yield relative amount of adsorbed/desorbed water as RH changes, while the FTIR O-H stretching band probes the total water content in the RSF samples.

We then allocated these peaks into two groups: the bound water with strong H-bonding (peaks at 3000 cm⁻¹, 3210 cm⁻¹ and 3290 cm⁻¹) and the mobile water with weaker H-bonding (peaks at 3410 cm⁻¹ and 3550 cm⁻¹)[46] (Fig. 4a, b). These allocations align with those reported in literatures[46,48,49,51] (see Supplementary Notes 1, 2 and Fig. S8b). We noted that, as RH increases from 10% to 90%, the amount of the bound water in Silk-H and Silk-M increases by 81.6% and 51.8%, respectively, compared to its initial bound water content at 10% RH, while the amount of the bound water in Silk-L only increases by 11.4% (Fig. 4c). It also suggests that the fraction of bound and mobile water is variable in Silk-H and Silk-M during hydration and dehydration, resulting in higher degrees of structural reconfigurations in the amorphous regions of Silk-H and Silk-M during hydration. However, the bound water structure remains fixed in Silk-L, likely attributed to the reduced domain size and the constraining of neighboring β-sheet domains, which limit the structural reconfigurations of amorphous domains. These differences could be attributed to the increasing H-bonding sites resulting from higher degrees of structural reconfigurations in the amorphous regions of Silk-H and Silk-M during hydration. To visualize the contribution of bound and mobile water to the total water exchange in RSF samples during hydration and dehydration, we also normalized the changes in bound and mobile water (compared to 10% RH) to the total water content of each RSF sample at 90% RH (Fig. 4d). The normalized changes in bound and mobile water clearly show that the mobile water dominates the water adsorption and desorption during hydration and dehydration, which is consistent with simulations in previous studies[24]. By comparing the total changes in water content obtained from the FTIR deconvolution to the total mass change measured by DVS as RH changes between 10% and 90%, we also estimated the bound water mass ($m_b$) to RSF protein dry mass ($m_p$) ratios ($m_b/m_p$) to 0.312, 0.318, 0.28 at 10% RH and 0.569, 0.48, and

0.318 at 90% RH for Silk-H, Silk-M, and Silk-L, respectively. These estimated values are comparable to those (0.2–0.3) reported for other proteins[52,53]. Using the bound and mobile water content deconvoluted from the FTIR spectra, we also plotted the B/M water ratios at different RH (Fig. 1c) and compared the results with the dehydration-induced stress (Fig. 1i). We found that all RSF samples start to exert force when the B/M ratio reaches a similar level (0.84 for Silk-H, 0.8 for Silk-M, and 0.88 for Silk-L) during dehydration. This critical B/M ratio also corresponds to the viscous-rubbery phase transitions of RSF samples (Fig. 1f–h) described in the Zener model. Our observations suggest that the critical viscous-rubbery phase transition for the RSF to exert force on external loads is likely governed by the B/M ratio of the confined water rather than secondary structures.

## Discussion

Our study shows a quantitative approach explaining macroscale dehydration-induced pressure in non-porous RSF through connections with water structure. This finding builds on previous results, showing that increased crystallinity and stiffness will enhance WR pressure and energy density of RSF[13,14]. The energy conversion from the chemical potential of water resulting in mechanical deformation likely occurs within non-crystalline hydrophilic regions, and the high mechanical stiffness resulting from increased crystallinity contributes to the effective transfer of the converted mechanical energy in the amorphous region to external loads. During dehydration, we observe that energy transfer from the chemical potential to macroscopic scale mechanical energy initiates when RSF samples undergo viscous-rubbery phase transitions governed by the B/M ratio of water confined in the WR material. As the B/M ratio reaches the lower boundary (0.8–0.9) during dehydration, the material can begin to generate pressure. However, we also noted that when the B/M water ratio in Silk-L is higher than 1.4–1.6 at low RH, the WR pressure plateaus. This could be due to the limits of RSF's structure in translating pressure to external loads. These findings suggest the dominant role that water structure plays in energy conversion during water-responsiveness. Our findings provide a strategic approach to controlling and predicting the stress generated during hydration and dehydration of biopolymers for diverse engineering applications.

## Methods

### Preparation of RSF solutions and thin films

RSF solutions and films were prepared using a previously reported method[13,16]. *Bombyx Mori* silkworm cocoons (TTSAM, Amazon) were cut into strips and boiled in a 0.02 M NaCO$_3$ (Sigma-Aldrich) solution for 30 min, followed by three rounds of rinsing and soaking (3x rinsing and 20 min soaking per round) in ultrapure water (Advantage A10, Milli-Q®). After a day of drying under ambient conditions, the degummed silk fibroin fibers were dissolved (0.25 g ml$^{-1}$) in a 9.3 M LiBr (Sigma-Aldrich) solution at 60 °C for 2 h. The solution was subsequently dialyzed in ultrapure water using regenerated cellulose dialysis tubing (MWCO 3500 Daltons, Fisherbrand™), with periodic changing of the water every 30 min for 2 h[13–15]. The dialyzed solution was then centrifuged at 7000 × *g* for 20 min to remove solid particulates. The resulting solution was refrigerated at 4 °C, with the maximum usable time window of 2 weeks after the dialysis step.

To prepare RSF films, the RSF solution was cast into a polystyrene petri dish (Fisherbrand™), degassed for 15 min under vacuum, and then left to dry at ambient conditions for a day. The dried film was peeled off from the petri dish and cut to form the Silk-H films. The Silk-M and Silk-L films were prepared by further treating the Silk-H films with water and methanol, respectively. Silk-M films were prepared by exposing the Silk-H films to high RH (>90%) at 45 °C for 24 h and subsequently drying them overnight at ambient conditions. Silk-L films were prepared by submerging the Silk-H films into 99.8% methanol (Sigma-Aldrich) for 1 min and allowing them to dry overnight at ambient conditions. Both water and methanol treatments enhance chain mobility and allow for the self-assembly of hydrophobic β-sheet nanocrystals, which is a more energy favorable configuration.

### RH control

The different RH levels were achieved by mixing dry N$_2$ gas with N$_2$ gas bubbled through a water-filled reservoir. The ratio of dry and humid gas was controlled by regulator valves, and the resulting RH was monitored using either a handheld RH probe (HM41, Vaisala) or a small-scale RH sensor (HIH-4021, Honeywell), with a data acquisition device (NI USB-6001 DAQ, National Instruments) and a LabVIEW program for data acquisition. Various environmental control chambers were used for the experiments in this study. The chamber used with FTIR placed the sample between two CaF$_2$ windows, perpendicular to the beam path, in the absorption measurement configuration. An acrylic RH control chamber, equipped with a calibrated RH sensor (HIH-4021, Honeywell), was used to control the RH levels during the WR stress, strain, and Young's modulus measurements. A dedicated acrylic chamber, with a quartz window was constructed for the RH control in the spectrophotometer experiment. All tests were performed under standard conditions of 293 K and 1 atm.

### WR strain measurement

The WR free strain of RSF films was obtained from the length changes of the free films under unstressed conditions, as RH decreases from 90% to 10% in 10% increments. After the samples were equilibrated at individual RH levels for 5 min, the length of the samples was recorded by a 40 M pixel camera and analyzed by an imaging processing software, ImageJ. The WR strain was calculated by normalizing the measured length at different RH to that at 10% RH during dehydration. Subsequently, a segmented linear regression, with Levenberg–Marquardt algorithm was used to fit the data with the strain at 10% fixed to zero. The data from the linear fit lines was used in the viscoelasticity analysis.

### WR stress measurement

The WR stress of RSF films was measured by using a Dynamic Mechanical Analysis (DMA) device (ElectroForce 5500, TA Instruments). RSF films of 10 mm × 2–4 mm × 0.01–0.09 mm were clamped

into acrylic sample holders, pre-tensioned to 0.01 N and equilibrated at 90% RH. During dehydration, the force generated by RSF films was recorded when the force reaches equilibrium at each RH, and the dehydration-induced pressure was subsequently calculated by dividing the force to the cross-sectional area of the RSF film. The measured non-zero points were fitted with a non-weighted linear fit model, with $R^2$ values of 0.54945−Silk-H, 0.94517−Silk-M and 0.90323−Silk-L. The inception RH value for each sample was obtained by dividing the negative intercept value by the slope of the dehydration induced stress. The linear fit data was used later in the viscoelasticity analysis.

### Young's modulus measurement

Young's moduli (unrelaxed elastic moduli) of RSF films were measured using a DMA instrument (ElectroForce 5500, TA Instruments). RSF films of 10 mm × 0.5–1.5 mm × 0.01–0.03 mm were clamped in place and exposed to the RH values for 15 min at initial condition (10% RH) and then 5 min between each RH step (10, 30, 50, 70, 90% RH) before measuring. A step function of strain (1% at a rate of 1% s$^{-1}$) with a relaxation time of 10−15 s was applied to RSF films. Subsequently, the RSF films were relaxed and reset for a total of 3 repeats, with 10–15 s between each stress/relaxation cycle. The strain was increased at high RH levels (70–90%) until a measurable stress signal was obtained. The Young's modulus was determined by performing a linear fit of the stress-strain curve in the elastic region.

### RSF/polyimide bilayers

We demonstrated the RSF samples' propensity to actuate at different RH levels by preparing RSF/polyimide bilayers. A 6 μm thick layer of RSF was deposited on 12 μm films of plasma-treated (30 s in 75% Ar:25% O2 plasma) polyimide and left to dry overnight. Subsequent samples were then exposed to their respective post-treatments. The sample size was 3 × 6 mm, with a weight of 10.6 mg attached to the end of the bilayer films.

### Viscoelasticity analysis of RSF films

The measured WR and mechanical behaviors of the RSF films were analyzed by the standard linear solid (Zener) model (Fig. 1f), where the dynamic response of stress and strain can be described by:

$$D_R\sigma + \eta(D_R - D_U)\frac{d(D_U\sigma)}{dt} = \varepsilon - \varepsilon_S + \eta(D_R - D_U)\frac{d(\varepsilon - \varepsilon_S)}{dt} \quad (1)$$

where the $D_U$ and $D_R$ are the unrelaxed and relaxed compliances which are inverse of unrelaxed ($1/E_U$) and relaxed ($1/E_R$) moduli, respectively, $\eta$ is the uniaxial viscosity, $\varepsilon$ is the strain of the film, $\varepsilon_S$ the free strain due to RH changes determined from the length change from the reference state described in Fig. 1d, and $\sigma$ the stress induced in the RSF films. At a certain RH level, under high strain rate (unrelaxed), Eq. (1) gives:

$$\sigma = E_U\varepsilon \quad (2)$$

For the WR free strain and dehydration-induced stress measurements, the effect of the dynamic component of the dashpot (uniaxial viscosity) can be disregarded, as steady states of RSF films are reached for each RH level (relaxation time >5 min). At non-stressed conditions ($\sigma = 0$), where the length change of the free-standing films is mainly due to RH changes, the strain measured equals the free WR strain, $\varepsilon = \varepsilon_S$. The measured WR strain is modeled by a segmented linear fit function (Fig. 1d). Similarly for the dehydration-induced stress measurements ($\varepsilon = 0$ and $\sigma = -E_R\varepsilon_S$), the linear regression of the non-zero measured data points, is extrapolated to the intersection with the $\sigma = 0$ (Fig. 1e). The relaxed modulus ($E_R$) at different RH is then obtained from the linear fits of the WR strain ($\varepsilon_S$) and dehydration-induced stress ($\sigma$) using $E_R = \Delta\sigma/-\Delta\varepsilon_S$ (3). As indicated in Fig. 1h, in the low RH regime, the RSF film behaves as a solid, and its elastic modulus has

weak dependence on the RH level. This corresponds to the actuating regime, where the material has the stiffness to exert the observed force. At high RH levels, due to the increased intake of water, the RSF films behave like a visco-fluid material.

## FTIR spectroscopy

An ATR-FTIR (IS50, Nicolet) setup was used to determine the secondary structures of RSF films using previously reported methods[38,54]. The spectrum was acquired from 800 to 4000/cm over 64 scans, with a resolution of 4/cm. After the measurement, the Amide I and II bands ranging from 1470 to 1725/cm were isolated, baseline subtracted, and smoothed prior to fitting (OriginPro 2019). Ten peaks were selected for the deconvolution of the Amide I and II bands based on peaks identified in previous studies[38,54]: 1510/cm (β-sheet), 1530/cm (Random coil), 1540/cm (α-helix), 1565/cm (β-turn), 1592/cm (non-baseline correction peak[38]), 1620/cm (inter-/intra-molecular β-sheet), 1640/cm (α-helix), 1655/cm (random coil), 1680 (β-turn), 1695/cm (intermolecular β-sheet). The deconvolution of the Amide I and II bands shows an agreement in the ratio between amorphous and crystalline structures (Fig. S1). The secondary structure quantification was obtained only from the deconvolution of the Amide I band. As the adopted deconvolution parameters could influence the estimated amounts of secondary structures (Fig. S2), we have grouped the secondary structures into crystalline domains (1620 and 1695/cm β-sheet peaks) and amorphous domains (1640, 1655 and 1680/cm peaks).

The H-bonding networks in RSF films were characterized with the FTIR of the O-H stretching band (Supplementary Notes 1 and 2). The FTIR absorption measurements were performed on thin (>10 μm) RSF film samples using the FTIR (Vertex 70, Bruker) spectrometer equipped with an RH control chamber with $CaF_2$ windows.

For the deconvolution of the OH-stretch peak, a single RH cycle (hydration/dehydration from 10% to 90% in 10% increments) is considered. The O-H stretching band, ranging from 2500 to 3700/cm, was acquired over 64 scans, with a resolution of 2/cm. The baseline correction, CH-stretch peak (2800–3100/cm) correction and smoothing of the signal is performed before the deconvolution (OriginPro 2019). Five peak positions (AAD−3000/cm, AADD−3210/cm, water bound to organic matter−3290/cm, AD−3410/cm and ADD−3550/cm), corresponding five different H-bonding scenarios[45,46,48–50], were determined by establishing a reasonable physical model to approximate the H-bond populations in the samples and ensuring a good fit (see Supplemental Note 2 and Fig. S8). The latter two are considered more loosely bound (mobile water), with one or two H-bonds per molecule, while the first three are associated with fully or close to tetrahedrally arranged water (bound water)[45,46,48,49]. Subsequently, these five peak positions were used for the Gaussian deconvolution of the O-H stretching band. The area of the deconvoluted peaks was used to estimate the populations of mobile and bound water. In Fig. 4c, the areas of the deconvoluted peaks are normalized to the total peak area of the initial 10% RH case (to 100%). For Fig. 4d, the relative water content changes in bound and mobile water were obtained by using the following formula:

$$\frac{A_X(RH) - A_{X,i}(10\%RH)}{A_T(RH) - A_{T,i}(10\%RH)} \times 100\ (\%) \qquad (3)$$

where $A_X(RH)$ represents the amount of bound or mobile water and $A_T(RH)$ represents the total amount of water at a given RH value, with the $A_{X,i}(10\% RH)$ and $A_{T,i}(10\% RH)$ indicating their initial values at 10% RH.

## SEM

To image the cross-section of RSF films, thin film samples were first submerged in liquid nitrogen for 30 s and cryo-fractured to expose their cross-sections. The samples were then coated with ~3 nm of platinum using a sputter coater (EM ACE600, Leica) and imaged using an Environmental SEM (Quattro S, Thermo Scientific).

## AFM

RSF's surface topographies were imaged by an AFM (Multimode 8, Bruker). RSF films were glued to silicon substrates with double-sided tape. The topographies were imaged by using an AFM probe with a tip radius of ~2 nm (SCANASYST-Fluid, Bruker) in tapping mode.

## DVS

The water sorption isotherms of the RSF films were measured using a DVS equipment instrument (DVS Intrinsic Plus, Surface Measurement Systems). The measurements were conducted at 25 °C, 1 atm, and a flow rate of 200 cm³/min, with a convergence criterion set at 0.01% of dm/dt. The measurements were performed for three RH cycles from 10% to 90% RH and back, in 10% RH increments.

## Fluorescence spectroscopy

Fluorescence data were acquired by a spectrofluorometer (FP-8500, Jasco) with a Xe lamp excitation source, through a 0.25 mm thick quartz cover slip window (Ted Pella). The emission/excitation maps were recorded at 10% and 90% RH. The excitation wavelength range was selected from 220 to 400 nm, with a 5 nm interval and the 1 nm excitation slit opening. The emission spectra were recorded from 280 to 550 nm with a 1 nm resolution, 200 nm/min scan speed and 2.5 nm slit opening. The intensity was normalized to the highest fluorescence signal on the emission/excitation map, with the excitation light omitted.

The 290 nm (±0.5 nm excitation slit) excitation wavelength for the emission graph was then selected wavelength in order to maximize tryptophan excitation and minimize the excitation of other aromatic residues, such as tyrosine, which are known not to redshift in the presence of polar solvents[43]. Also, due to resonant energy transfer overlap between tyrosine and tryptophan, any minimal tyrosine excitation will tend to be transferred efficiently to tryptophan excitation[55], assuring the dominant tryptophan emission at this excitation wavelength. The measurement range was 295–550 nm, with 0.1 nm resolution, 200 nm/min scan speed, and an emission slit opening of 2.5 nm. Each spectrum was averaged over two accumulations. Emission peak maximum analysis was performed on individual spectra acquired during three RH cycles (in 10% increments, from 10% to 90% RH), with a minimum of 10 min relaxation time for each equilibrium position. The spectra were smoothed using the Adjacent-Averaging method (over 50 datapoints) to remove the noise, normalized, and determined the peak maximum (OriginPro 2019) using the second-order derivative method. The measured spectra were normalized to the highest value, and only spectral shifts (not intensity data) were considered for analysis.

## Bound water to non-hydrated protein mass ratios

The FTIR deconvolution peak area change corresponds ($\Delta A_{10-90\%RH}$) to the proportional loss of mass in the DVS measurements between 90 and 10% RH ($\Delta m_{10-90\%RH}$). The water content at 10% RH detected by the FTIR was then subtracted from the total mass measured in the DVS measurements to determine the dry protein mass ($m_p$). The mass of the remaining water was multiplied by bound-to-total (B/T) water ratio to get the remaining water mass fraction of the bound water at 10% and 90% RH ($m_b$). The $m_b/m_p$ ratio gives us the gram per gram approximation of the protein hydration shell or bound water, often used in protein characterization[20,53].

## Reporting summary

Further information on research design is available in the Nature Portfolio Reporting Summary linked to this article.

## Data availability

The data generated in this study are provided in the Supplementary Information/Source Data file. Source data are provided with this paper.

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

## Acknowledgements

This work was supported by the Air Force Office of Scientific Research (AFOSR) (FA9550-21-1-0144 to R.S.T. and X.C.) and the National Science Foundation (NSF) (CBET-2238129 to X.C. and ITE-2344305 to X.C. and R.S.T.). We would like to thank Dr. Elizabeth Biddinger and Dr. Rein Ulijn for the use of FTIR and spectrophotometer.

## Author contributions

X.C. and R.S.T. conceived and initiated the project. D.P., Y.J. and M.K. prepared the RSF. M.K. acquired the SEM images. D.P. performed DMA, FTIR, DVS, and fluorescence measurements and analyzed the data. H.Y. modeled the WR and mechanical properties. All authors contributed to data analysis and discussed the results. X.C., D.P. and R.S.T. wrote the paper, and X.C. and R.S.T. supervised the project.

## Competing interests

The authors declare no competing interests.
