## [Peer Review File · Nature Communications]

The role of water mobility on water-responsive actuation of silkREVIEWER COMMENTS

Reviewer #1 (Remarks to the Author):

The manuscript describes an experimental study that investigates how relative humidity (RH) affects the water response (WR) of three types of regenerated silk fibroin (RSF) films: Silk-H, Silk-M, and Silk-L. The same group has previously provided detailed macroscopic characterization of these materials. In this study, they aim to explore the microscopic rationale behind their results. The researchers establish an interesting relationship between micro and macro properties, which I believe is significant enough to be published.

The three protocols differ in the post-dry treatment of the films: no treatment for Silk-H; long exposure to high RH at high temperature and subsequent dry for Silk-M; submersion into methanol and dry for Silk-L.

The samples vary in water adsorption. Silk-H, Silk-M, and Silk-L adsorb high, medium, and low water content, respectively (e.g., see Fig. 2d, showing the relative mass change of each sample).

They also differ in WR actuation energy densities once they undergo RH cycles: less in Silk-H, intermediate in Silk-M, and more in Silk-L.

Here, the authors refine of the macroscopic characterization by measuring changes in strain, stress, and elastic modulus as a function of RH. They find that the WR in stress, and the RH threshold value of the response, is larger in Silk-L compared to Silk-M, which in turn is larger than in Silk-H. This same relationship holds for the response in elastic modulus, indicating the viscous-to-rubbery phase transitions in the films. Therefore, as the water content increases, the WR decreases, which is consistent with the previous energy density results.

To determine the ratio of bound (B) to mobile (M) water, they analyzed the Fourier transform infrared (FTIR) spectra of the samples within the OH stretching wavenumber region. By deconvoluting the spectra, they calculated that the B/M water ratio is highest in Silk-L, followed by Silk-M, and lowest in Silk-H.

According to their findings (Fig. 1i), the stress response of the Silk-H, Silk-M, and Silk-L samples collapse into a single master curve when plotted based on the B/M water ratio. They conclude that a critical value, estimated to be around 0.8-0.9, of the B/M water ratio is responsible for controlling the WR of the samples and the phase transition from a viscous to a rubbery state.

To understand where water is absorbed, they refine their published analysis [13], confirming that Silk-L has more β -Sheet nanocrystals than in Silk-M, which in turn has more than Silk-H (Fig. 2a-c). Using the tryptophan fluorescence shift during the RH cycle, they find that Silk-H experiences a larger change in adsorbed water than Silk-M, which in turn has a larger change than Silk-L (Fig. 3c).

Since tryptophan is mainly present in the amorphous (disordered) regions of RSF, they conclude that Silk-H amorphous regions undergo larger conformation changes than Silk-M, and small changes for Silk-L.

They interpret the collapse of the tryptophan fluorescence shifts maxima onto a single curve as a function of the adsorbed water content as evidence that the adsorption is localized within the RSF hydrophilic amorphous regions.

Based on my understanding of the manuscript, the key findings are illustrated in Fig.1i and Fig.3d. The first collapse demonstrates that the ratio of B/M water plays a crucial role in determining the macroscopic properties. The second collapse reveals that the adsorption of water in the hydrophilic and unstructured regions is directly proportional to the total water content.

While it is expected that the adsorbed water goes mainly to the hydrophilic regions, the first result is intriguing. The existence of a critical B/M water value that determines the macroscopic properties of films, regardless of the proteins' microscopic structure, offers a strategy for material design in high-performance actuators.

However, before recommending the publication in a high-impact journal, I have a few suggestions and comments based on my expertise and research. For this reason, I decided to sign my reviewer report.

1) The description of all nanoconfined water as consisting of bound and mobile water is accurate, provided that it specifies that the mobile component includes both bulk-like and unbound hydration water. Our molecular dynamics simulations of membranes at various hydration levels have demonstrated this, as shown in [<https://doi.org/10.1016/j.molliq.2018.10.074>]. Bound water refers to water that is within short distances (<1 nm) from the membrane. It is slow and distinct from bulk water [<https://doi.org/10.1021/acsnano.0c02984>]. Bound water tightly interacts with biomolecules, often bridging different hydrophilic parts, which contributes to the structural stability, and never leaves, even at very low RH. In fact, it is part of the bio-surface and forms an interface with the unbound hydration water [<https://doi.org/10.1016/j.molliq.2018.10.074>]. This is consistent with the present analysis, which shows that water is bound to the Silk-L even at very low RH and contributes to their mechanical properties.

We also found that the amount of bound and unbound hydration water is always relevant because they extend as far as 2.5 nm from the membrane [<http://dx.doi.org/10.1007/s11467-017-0704-8>], redefining the concept of hydration water near bio-interfaces [<https://doi.org/10.1116/6.0000819>].

In particular, bound water slowly evolves and forms one H-bond (HB) with the biomolecule [<https://doi.org/10.1016/j.molliq.2018.10.074>] and 2 or more HBs with other water molecules [black symbols in Fig.4 of <https://doi.org/10.1021/acsnano.0c02984>]. Hence, it contributes to the strong component in the deconvolution of the FTIR spectra.

On the other hand, unbound hydration water has weaker HBs (DA and DDA) [red symbols in Fig.4 of <https://doi.org/10.1021/acsnano.0c02984>], lacks water-biomolecule HBs [Fig. 3

<https://doi.org/10.1016/j.molliq.2018.10.074>], and is more mobile than bound water [<https://doi.org/10.1016/j.molliq.2018.10.074>]. Therefore, it contributes to the 'mobile' water component considered in the present manuscript.

Overall, all these results support those presented here, and discussing them in the manuscript would strengthen it.

2) It would be also worth citing in the introduction that bound water plays a crucial role in achieving a consistent thermodynamic understanding of the instability of structured proteins at low temperatures and high pressure [<http://link.aps.org/doi/10.1103/PhysRevLett.115.108101>], transferable protein design [<http://dx.doi.org/10.1103/PhysRevX.7.021047>], and free-energy barriers toward protein aggregation [<https://doi.org/10.1002/cphc.201900904>]. Additionally, it promotes water-mediated protein adsorption onto hydrophobic surfaces [Fig. 5c-f in <https://doi.org/10.1021/acs.jpcc.3c00937>] due to the unusual free energy of hydrophobic nanoconfined water double-layers that result from the properties of hydrogen bonds [<https://doi.org/10.1021/acsnano.1c07381>]. This can lead to hydration pressure as high as 1 GPa when confinement is below 1 nm [Fig. 7 in <https://doi.org/10.1016/j.molliq.2020.114027>], consistent with experiments [<http://dx.doi.org/10.1038/ncomms12168>].

3) The authors discuss the interpretation of attenuated total reflection (ATR) FTIR spectroscopy to estimate the conformational changes of the RSF proteins in each sample. They refer to their published analysis [13] on the amount of β -Sheet nanocrystals and Refs. [33, 47]. However, they should mention that the estimate of different amounts of structures is influenced by the adopted deconvolution parameters [<https://doi.org/10.3109/10409239509085140>].

To support this deconvolution, complementary measurements are necessary, especially when the analysis includes five different structures, as shown in Fig. 2, without accounting for the strong overlap of the bands assigned to different secondary structures (e.g., disordered structures and α -helical) [<https://doi.org/10.1007/s00249-021-01502-y>].

Furthermore, it's unclear to me whether Fig. 2 is necessary for this work or not. While it is a refinement of Fig. 2 in Ref. [13], it does not seem essential in the present work. If it is relevant, it should be clarified better why within the manuscript. If instead it is presented to support that Silk-L has more β -Sheet nanocrystals, this point has already been discussed in Ref. [13] and this figure could be removed also in light of the above-mentioned deconvolution issues.

On the other hand, it would be enough to note that methanol enhances secondary structures and destabilizes the tertiary structure, generating a "molten globule"-like state [<https://doi.org/10.1038/srep19500>]. This reduces the exposure of the protein surface to hydration and, as a consequence, the sample's capability to adsorb water, which is consistent with the characterization of Silk-L.

Also, Silk-M samples' reduced capability to adsorb water is likely due to their exposure to high relative humidity at high temperatures, which destabilizes the proteins and causes misfolding. This,

in turn, increases the molten globule state, leading to reduced exposure of more hydrophilic domains. As a result, the detailed analysis presented in Fig. 2a-c may not be necessary for the essential discussion and could be removed or moved to the supplementary information.

4) In a similar fashion, should Fig. 2d be included in the supplementary information to demonstrate consistency with previous works, or has it already been published elsewhere?

5) On page 10, the characterization of H-bonding networks by FTIR is discussed. The terms 'high H-bond density' and 'lower H-bond density' water are used to refer to tightly bound water and bulk-like mobile water respectively. However, it is unclear to me whether these terms can be used interchangeably with strong and weak H-bonds. Indeed, the density of H-bonds in water is not necessarily related to the density of water-biomolecule H-bonds, as shown in Figure 3 of [<https://doi.org/10.1016/j.molliq.2018.10.074>]. Therefore, it is recommended that the use of these expressions be either removed or discussed in greater detail with supporting evidence.

6) In the same section, the O-H stretching band in FTIR is divided into five populations of H-bonds. However, is this really necessary? The objective is to estimate only the amount of bound and mobile water, which means only two populations are required.

For example, only two populations (called 'ice-like' and 'liquid-like') are used in the reference cited in the supplementary information, Ref.[16], for increasing RH, although an intermediate population ('transitional') is discussed for intermediate RH. This deconvolution is consistent with the above discussion of bound, unbound, and bulk-like water.

In another example (water confined in micelles) a three-component analysis gives an excellent fit to the data with a minimum number of parameters [<https://doi.org/10.1063/1.1894929>]. Again, this approach is consistent with the bound, unbound, and bulk-like water populations near biomolecules, described above. A deconvolution with three (or possibly two) populations would be sufficient to estimate the B/M water ratio. Additionally, it would be consistent with the ab initio molecular dynamics analysis we conducted for bulk water (Fig. 11 in [<https://doi.org/10.1016/j.molliq.2022.119936>]). In any case, it should be mentioned that the deconvolution with Gaussians is not unique [<https://doi.org/10.1063/1.1894929>].

Moreover, it appears that the five populations' deconvolution is not entirely consistent with the method used in Ref.[38]. In the reference, the fifth population is defined as free-OH, whereas here it is defined as 'water-solid' and associated with a different peak value. To maintain consistency, there should be six populations, including free-OH. It is important to note that the free-OH population is replaced by five-coordinated water when detailed HB-network analysis is performed on bulk water (as shown by the orange triangles in Fig.4 of [<https://doi.org/10.1021/acsnano.0c02984>]).

Furthermore, in Ref. [43], only two populations are used to support the 'water-solid' choice: water-air interface for strong hydrogen bonds near the hydrophobic interface and fiber-water interface for weaker hydrogen bonds near the hydrophilic interface.

Therefore, how robust is the interpretation of the results presented here if only three populations were used to deconvolute the O-H stretching band (Fig. 4, S3-S5)?

7) The sentence "the amount of the bound water in Silk-H and Silk-M increases with rising RH, while in Silk-L, the amount of the bound water remains relatively constant across the entire RH range from 10% to 90% (Fig. 4c)" appears to contradict Fig. 4d, where the changes in bound water for all three samples are comparable. Is it possible that the authors meant to refer to the structural changes shown in Fig. 3c, which are less noticeable in Silk-L compared to the other two samples?

This would be more consistent with the subsequent sentence "These differences could be attributed to the increasing H-bonding sites resulting from higher degrees of structural reconfigurations in the amorphous regions of Silk-H and Silk-M during hydration".

However, the authors should also mention that the difference in Fig.3c could be associated with the saturation of the reduced amount of hydrophilic regions exposed to water after the methanol treatment, independent of the RH.

8) They should also clarify why the methanol treatment should limit the 'structural reconfiguration in the amorphous regions'? Is it because it favors molten globule structures?

9) Furthermore, it should be noted in the manuscript that the fact "that the mobile water dominates the water adsorption and desorption during hydration and dehydration" is consistent with the molecular dynamics calculations performed in [<https://doi.org/10.1016/j.molliq.2018.10.074>] which show that the bound water population is the least affected by dehydration (Fig. 2).

10) The statement "We localize the water exchange predominantly in the hydrophilic amorphous region" and its reiteration in "the B/M ratio of water confined in the amorphous region" cannot be solely based on the present results. As discussed earlier, the collapse in Fig.3d only indicates that water adsorption in the hydrophilic unstructured regions is proportional to the total water content, which is expected. There is no concrete evidence that water predominantly adsorbs in the disordered regions. In principle, water can adsorb in any hydrophilic region, irrespective of its structure. Hence, these sentences need to be rephrased.

11) According to their observations, the film WR stress possibly saturates when the B/M water ratio exceeds 1.4-1.6, as shown in Figure 1i. They interpret this as being likely due to the maximal mobile water capacity of the unstructured domains. However, this point requires further explanation. It is worth noting that these high values of B/M water ratio are only accessible for the Silk-L sample at the lowest RH, where most water is bound. Under these conditions, any further decrease in RH would not decrease the bound water, which is consistent with [<https://doi.org/10.1016/j.molliq.2018.10.074>]. Therefore, the mechanical response would remain unchanged.

12) Minor points:

a) On page 7, in the section titled "Dehydration-induced strain and stress of RSF", there is a typo in the last two lines where Silk-H and Silk-L are interchanged. The same error is also present in the fifth and fourth lines before the last, at the end of the same section on page 8.

b) DVS stands for dynamic vapor sorption, not dynamic water sorption, as the manuscript uses.

c) The protein regions described as amorphous (unstructured) in this manuscript are commonly referred to as intrinsically disordered regions in protein literature [e.g., [https://doi.org/10.1002/1097-0134\(20010101\)42:1>38::AID-PROT50<3.0.CO;2-3](https://doi.org/10.1002/1097-0134(20010101)42:1>38::AID-PROT50<3.0.CO;2-3)]. Clarifying their synonyms would help connect them with other subfields in protein science.

d) In section "Viscoelasticity analysis of RSF films" the symbol 'd' in Eq.(2) is not defined. The Eq.(2) should be on a separate line.

e) Due to the reasons explained above, they should avoid referring to their convolutions of "FTIR of the OH stretching band, as it is a well-established method for studying H-bonding of water in materials." Instead, they should cite the references they follow with the above-mentioned caveats.

f) The choices of peak values on page 18 for AAD, ADD, and AD do not coincide with those reported in the supporting Ref.[38].

g) In Supplementary Notes, section 1, 'analysis as was' should be 'analysis was'.

To sum up, the outcome presented in Fig. 1i is quite intriguing as it provides us with a better understanding of the mechanisms regulating the WR of these films. This finding could be relevant in the field of new high-performance actuators. However, the authors are advised to follow the guidelines mentioned above to make the message of this manuscript more robust and clear.

--

Dr. Giancarlo Franzese, Profesor Titular de Universidad en Física
de la Materia Condensada

Reviewer #2 (Remarks to the Author):

In this work, the authors prepared three types of regenerated silk fibroin proteins to investigate how nanoscale water limitation controls macroscale dehydration-induced actuation in nonporous/nanoporous RSF. It indicates that the increased crystallinities and stiffness will enhance WR pressure and energy density of RSF. However, there are still some issues that have not been well resolved, so I do not recommend the publication of this work. The following issues need

to be addressed:

1. In figure 3b, the authors suggest that all RSF samples exhibit a dramatic shift in their tryptophan emission wavelength when RH changes between 10% and 90%. However, neither Silk-M nor Silk-L shows obvious changes and differences, please explain this reason in detail.
2. In figure 3d, the authors suggest that adsorption/desorption is mainly located in the amorphous regions of RSF samples. Please explain this reason in detail.
3. In figure 2a, the authors prepared two other films by treating Silk-H films with water and methanol, and realized the modification of the secondary structure of silk fibroin protein. Such operation seems very simple, please explain this mechanism from the molecular level.
4. In figure 2d, low hysteresis of the measured water sorption isotherms suggests that all RSF films share similar nonporous/nanoporous solid structures. Please add a detailed discussion and characterization of this conclusion.
5. This work lacks the application work of materials, and should be supplemented with reasonable and novel application displays, so as to more intuitively and strongly confirm the mechanism study of RSF microstructure in this paper.
6. In Figure 2, ATR-FTIR spectra were applied to characterize the secondary structure of the RSF sample before and after water absorption. The characterization of secondary structures is essential for mechanism interpretation, but the structural characterization in this paper is not persuasive and obvious enough. The evolution of secondary structure could be illustrated by more visual structural characterization methods such as SEM, AFM or SAXS/ WAXS.
7. This work investigates the macroscale dehydration-induced actuation in nonporous/nanoporous RSF and the effect of B/M ratio on structure. There are many similar studies. Please explain the fundamental innovation that distinguishes this study from other studies.

Reviewer #3 (Remarks to the Author):

This manuscript attempts to illustrate the underlying mechanisms of deformation in response to relative humidity of biological water-responsive materials. They found that the nanoscale confinement of water dominates the macroscopic dehydration-induced stress of the regenerated silk fibroin. Silk samples start to exert force as they undergo phase transitions from a viscous to rubbery state. These transitions occur when the bound-to-mobile (B/M) ratio of confined water reaches the same level. This critical B/M water ratio suggests a common threshold above which the chemical potential of water instigates the work of actuation in silk. Their findings may be beneficial to develop new physics to describe the WR behavior of biopolymers through confined water. They've done a very professional work, and reasonably illustrated the underlying mechanisms of deformation in response to relative humidity of biological water-responsive materials. However, I don't think their achievement is enough to be published in Nat. Commun.

1. This work revolves around the underlying mechanisms of deformation in response to relative humidity of biological water-responsive materials. A great deal of characterization and analytical

work was used to reveal the mechanism. However, in my opinion, water-responsive actuation of silk is not much different from other common water-responsive materials such as hydrogels and elastomers. The driving force of deformation mainly attribute to the uneven swelling from uneven distribution of water. Therefore, I can't find any important advance or novel theory in the manuscript.

2. The authors say their findings can act as guidelines for predicting and engineering silk's WR behavior, however, it's not proven that how can effective regulation silk's WR behavior mode based on their findings.

In a word, I don't deny that this is meaningful work, but it is not enough published in this journal.

Reviewer #4 (Remarks to the Author):

Reviewer #1:

The manuscript describes an experimental study that investigates how relative humidity (RH) affects the water response (WR) of three types of regenerated silk fibroin (RSF) films: Silk-H, Silk-M, and Silk-L. The same group has previously provided detailed macroscopic characterization of these materials. In this study, they aim to explore the microscopic rationale behind their results. The researchers establish an interesting relationship between micro and macro properties, which I believe is significant enough to be published.

We thank the reviewer for acknowledging the significance of this work and for the positive feedback.

The three protocols differ in the post-dry treatment of the films: no treatment for Silk-H; long exposure to high RH at high temperature and subsequent dry for Silk-M; submersion into methanol and dry for Silk-L.

The samples vary in water adsorption. Silk-H, Silk-M, and Silk-L adsorb high, medium, and low water content, respectively (e.g., see Fig. 2d, showing the relative mass change of each sample).

They also differ in WR actuation energy densities once they undergo RH cycles: less in Silk-H, intermediate in Silk-M, and more in Silk-L.

Here, the authors refine of the macroscopic characterization by measuring changes in strain, stress, and elastic modulus as a function of RH. They find that the WR in stress, and the RH threshold value of the response, is larger in Silk-L compared to Silk-M, which in turn is larger than in Silk-H. This same relationship holds for the response in elastic modulus, indicating the viscous-to-rubbery phase transitions in the films. Therefore, as the water content increases, the WR decreases, which is consistent with the previous energy density results.

To determine the ratio of bound (B) to mobile (M) water, they analyzed the Fourier transform infrared (FTIR) spectra of the samples within the OH stretching wavenumber region. By deconvoluting the spectra, they calculated that the B/M water ratio is highest in Silk-L, followed by Silk-M, and lowest in Silk-H.

According to their findings (Fig. 1i), the stress response of the Silk-H, Silk-M, and Silk-L samples collapse into a single master curve when plotted based on the B/M water ratio. They conclude that a critical value, estimated to be around 0.8-0.9, of the B/M water ratio is responsible for controlling the WR of the samples and the phase transition from a viscous to a rubbery state.

To understand where water is absorbed, they refine their published analysis [13], confirming that Silk-L has more β -Sheet nanocrystals than in Silk-M, which in turn has more than Silk-H (Fig. 2a-c). Using the tryptophan fluorescence shift during the RH cycle, they find that Silk-H experiences a larger change in adsorbed water than Silk-M, which in turn has a larger change than Silk-L (Fig. 3c). Since tryptophan is mainly present in the amorphous (disordered) regions of RSF, they conclude that Silk-H amorphous regions undergo larger conformation changes than Silk-M, and small changes for Silk-L.

They interpret the collapse of the tryptophan fluorescence shifts maxima onto a single curve as a function of the adsorbed water content as evidence that the adsorption is localized within the RSF hydrophilic amorphous regions.

Based on my understanding of the manuscript, the key findings are illustrated in Fig.1i and Fig.3d. The first collapse demonstrates that the ratio of B/M water plays a crucial role in determining the macroscopic properties. The second collapse reveals that the adsorption of water in the hydrophilic and unstructured regions is directly proportional to the total water content.

While it is expected that the adsorbed water goes mainly to the hydrophilic regions, the first result is intriguing. The existence of a critical B/M water value that determines the macroscopic properties of films, regardless of the proteins' microscopic structure, offers a strategy for material design in high-performance actuators.

However, before recommending the publication in a high-impact journal, I have a few suggestions and comments based on my expertise and research. For this reason, I decided to sign my reviewer report.

We thank the reviewer for the comments and suggestions that helped to further strengthen the manuscript. We have addressed the reviewer's comments below.

1) The description of all nanoconfined water as consisting of bound and mobile water is accurate, provided that it specifies that the mobile component includes both bulk-like and unbound hydration water. Our molecular dynamics simulations of membranes at various hydration levels have demonstrated this, as shown in [\[https://doi.org/10.1016/j.molliq.2018.10.074\]](https://doi.org/10.1016/j.molliq.2018.10.074). Bound water refers to water that is within short distances (<1 nm) from the membrane. It is slow and distinct from bulk water [\[https://doi.org/10.1021/acsnano.0c02984\]](https://doi.org/10.1021/acsnano.0c02984). Bound water tightly interacts with biomolecules, often bridging different hydrophilic parts, which contributes to the structural stability, and never leaves, even at very low RH. In fact, it is part of the bio-surface and forms an interface with the unbound hydration water [\[https://doi.org/10.1016/j.molliq.2018.10.074\]](https://doi.org/10.1016/j.molliq.2018.10.074). This is consistent with the present analysis, which shows that water is bound to the Silk-L even at very low RH and contributes to their mechanical properties.

We also found that the amount of bound and unbound hydration water is always relevant because they extend as far as 2.5 nm from the membrane [\[http://dx.doi.org/10.1007/s11467-017-0704-8\]](http://dx.doi.org/10.1007/s11467-017-0704-8), redefining the concept of hydration water near bio-interfaces [\[https://doi.org/10.1116/6.0000819\]](https://doi.org/10.1116/6.0000819).

In particular, bound water slowly evolves and forms one H-bond (HB) with the biomolecule [\[https://doi.org/10.1016/j.molliq.2018.10.074\]](https://doi.org/10.1016/j.molliq.2018.10.074) and 2 or more HBs with other water molecules [black symbols in Fig.4 of <https://doi.org/10.1021/acsnano.0c02984>]. Hence, it contributes to the strong component in the deconvolution of the FTIR spectra.

On the other hand, unbound hydration water has weaker HBs (DA and DDA) [red symbols in

Fig. 4 of <https://doi.org/10.1021/acsnano.0c02984>], lacks water-biomolecule HBs [Fig. 3 <https://doi.org/10.1016/j.molliq.2018.10.074>], and is more mobile than bound water [<https://doi.org/10.1016/j.molliq.2018.10.074>]. Therefore, it contributes to the 'mobile' water component considered in the present manuscript.

Overall, all these results support those presented here, and discussing them in the manuscript would strengthen it.

We concur with the reviewer's definition of bound and mobile water. We also appreciate the reviewer's detailed explanations connecting the simulations of water populations at various hydration levels to interactions with biomolecular surfaces. We have included additional discussions and references on these previous studies in the main text and Supplementary Information.

Changes made to the Introduction of the main text:

“While definitions vary, these confined water molecules are often referred to as “bound” water, meaning water molecules that are within short distances and interact strongly with the material's surface^{19,20}, as opposed to “mobile” water, that has little or no interaction with the material²¹. Studies have shown that the amounts of bound and mobile water are always relevant²³. In the hydration shell, bound water slowly evolves and forms H-bonds with biomolecules and other water molecules, while mobile water forms weaker H-bonds and lacks interactions with biomolecules²⁴. These hydration shells of bound and mobile water were found to be critical for structure stability^{24,25} and bio-functionality¹⁹, such as enzyme functionality²⁰, anti-freeze protein activity²⁶ and structural reconfigurations^{27–31}.”

Changes made to the References and Notes:

“23. Martelli, F., Ko, H.-Y., Borallo, C. C. & Franzese, G. Structural properties of water confined by phospholipid membranes. *Front. Phys.* 13, 136801 (2017).
24. Calero, C. & Franzese, G. Membranes with different hydration levels: The interface between bound and unbound hydration water. *Journal of Molecular Liquids* 273, 488–496 (2019).
25. Martelli, F., Crain, J. & Franzese, G. Network Topology in Water Nanoconfined between Phospholipid Membranes. *ACS Nano* 14, 8616–8623 (2020).
28. Cheung, M. S., García, A. E. & Onuchic, J. N. Protein folding mediated by solvation: Water expulsion and formation of the hydrophobic core occur after the structural collapse. *Proceedings of the National Academy of Sciences* 99, 685–690 (2002).”

Changes made to the Supplementary Information:

Supplementary Notes:

“Nanoconfined water can be grouped into bound and mobile water. Bound water refers to water within short distances and with strong interactions with the material's surface, while mobile water includes both bulk-like and unbound hydration water^{1–4}.”

Supplementary Information references:

- “1. Asakura, T. *et al.* Characterization of water in hydrated *Bombyx mori* silk fibroin fiber and films by ²H NMR relaxation and ¹³C solid state NMR. *Acta Biomater.* **50**, 322–333 (2017).
3. Chen, S.-H. *et al.* Observation of fragile-to-strong dynamic crossover in protein hydration water. *Proc. Natl. Acad. Sci.* **103**, 9012–9016 (2006).
4. Calero, C. & Franzese, G. Membranes with different hydration levels: The interface between bound and unbound hydration water. *J. Mol. Liq.* **273**, 488–496 (2019).”

2) *It would be also worth citing in the introduction that bound water plays a crucial role in achieving a consistent thermodynamic understanding of the instability of structured proteins at low temperatures and high pressure [<http://link.aps.org/doi/10.1103/PhysRevLett.115.108101>], transferable protein design [<http://dx.doi.org/10.1103/PhysRevX.7.021047>], and free-energy barriers toward protein aggregation [<https://doi.org/10.1002/cphc.201900904>]. Additionally, it promotes water-mediated protein adsorption onto hydrophobic surfaces [Fig. 5c-f in <https://doi.org/10.1021/acs.jpcc.3c00937>] due to the unusual free energy of hydrophobic nanoconfined water double-layers that result from the properties of hydrogen bonds [<https://doi.org/10.1021/acsnano.1c07381>]. This can lead to hydration pressure as high as 1 GPa when confinement is below 1 nm [Fig. 7 in <https://doi.org/10.1016/j.molliq.2020.114027>], consistent with experiments [<http://dx.doi.org/10.1038/ncomms12168>].*

We appreciate the reviewer’s suggestions, and we agree that the citations help give our narrative added depth. As suggested by the reviewer, we have included additional discussions and references in the main text.

Changes made to the Introduction of the main text:

“Specifically, bound water plays a crucial role in achieving a consistent thermodynamic understanding of the instability of structured proteins³⁰, transferable protein design²⁹, and free-energy barriers toward protein aggregation³¹.”

Changes made to the References and Notes:

- “29. Bianco, V., Franzese, G., Dellago, C. & Coluzza, I. Role of Water in the Selection of Stable Proteins at Ambient and Extreme Thermodynamic Conditions. *Phys. Rev. X* **7**, 021047 (2017).
30. Bianco, V. & Franzese, G. Contribution of Water to Pressure and Cold Denaturation of Proteins. *Phys. Rev. Lett.* **115**, 108101 (2015).
31. Bianco, V., Franzese, G. & Coluzza, I. In Silico Evidence That Protein Unfolding is a Precursor of Protein Aggregation. *ChemPhysChem* **21**, 377–384 (2020).”

3) *The authors discuss the interpretation of attenuated total reflection (ATR) FTIR spectroscopy to estimate the conformational changes of the RSF proteins in each sample. They refer to their published analysis [13] on the amount of β -Sheet nanocrystals and Refs. [33, 47]. However, they should mention that the estimate of different amounts of structures is influenced by the adopted deconvolution parameters [<https://doi.org/10.3109/10409239509085140>].*

To support this deconvolution, complementary measurements are necessary, especially when the analysis includes five different structures, as shown in Fig. 2, without accounting for the strong overlap of the bands assigned to different secondary structures (e.g., disordered structures and α -helical) [<https://doi.org/10.1007/s00249-021-01502-y>].

Furthermore, it's unclear to me whether Fig. 2 is necessary for this work or not. While it is a refinement of Fig. 2 in Ref. [13], it does not seem essential in the present work. If it is relevant, it should be clarified better why within the manuscript. If instead it is presented to support that Silk-L has more β -Sheet nanocrystals, this point has already been discussed in Ref. [13] and this figure could be removed also in light of the above-mentioned deconvolution issues.

On the other hand, it would be enough to note that methanol enhances secondary structures and destabilizes the tertiary structure, generating a "molten globule"-like state [<https://doi.org/10.1038/srep19500>]. This reduces the exposure of the protein surface to hydration and, as a consequence, the sample's capability to adsorb water, which is consistent with the characterization of Silk-L.

Also, Silk-M samples' reduced capability to adsorb water is likely due to their exposure to high relative humidity at high temperatures, which destabilizes the proteins and causes misfolding. This, in turn, increases the molten globule state, leading to reduced exposure of more hydrophilic domains. As a result, the detailed analysis presented in Fig. 2a-c may not be necessary for the essential discussion and could be removed or moved to the supplementary information.

We agree with the reviewer that the refinement itself is not essential in this work. However, we feel that it is worth showing the dramatic structural differences between these three types of RSF samples. Despite these structural differences, the similar bound-to-mobile water ratio at which all RSF starts to exert force highlight the dominant role of water structure in water-responsive actuation rather than RSF's secondary structures.

We also agree with the reviewer that the deconvolution parameters might affect the estimated amount of secondary structures. Given that the β -sheet is better predicted in non-deuterated conditions, as discussed in the suggested literature (De Meutter & Goormaghtigh, *European Biophysical Journal* (2021) - <https://doi.org/10.1007/s00249-021-01502-y>), we have revised Fig. 2b and the main text to only discuss the β -sheet crystalline and amorphous domains in these three RSF samples. In addition, we have examined the impact of the full width at half maximum (FWHM) on the estimated crystallinity (Fig. S2). While the lower limit of FWHM is fixed at 10/cm and the upper limits vary from 20/cm to 35/cm, the variation in the estimated crystallinity is less than 5% (see Fig. S2). As suggested by the reviewer, we have included additional discussions about the potential influence of deconvolution parameters on the estimated amounts of secondary structures in the Methods and Supplementary Information.

We also thank the reviewer for bringing up the interesting point of the formation of a "molten globule"-like structure during solvent treatment. The reduction in the exposure of the protein

surface to hydration and the destabilization of tertiary structure is the subject of ongoing experiments in our labs.

Changes made to the Methods:

“As the adopted deconvolution parameters could influence the estimated amounts of secondary structures (Fig. S2), we have grouped the secondary structures into crystalline domains (1620 and 1695/cm β -sheet peaks) and amorphous domains (1640, 1655 and 1680/cm peaks).”

Changes made to the Fig. 2:

Fig. 2 | Secondary structure, SEM and water sorption isotherms of the RSF samples. a, Normalized ATR-FTIR spectra of Amide I (1650/cm peak) and II (1520/cm peak) bands and the deconvolution peaks for Silk-H, Silk-M, and Silk-L samples. The deconvolution peaks are centered at 1510/cm (β -sheet), 1530/cm (Random coil), 1540/cm (α -helix), 1565/cm (β -turn), for Amide II peak, 1592/cm (baseline correction – see methods section) and 1620/cm (β -sheet – inter/intra-molecular), 1640/cm (α -helix), 1655/cm (Random Coil), 1680 (β -turn), 1695/cm (β -sheet – inter-molecular) for the Amide I peak. **b**, Crystalline and amorphous fractions of Silk-H, Silk-M, and Silk-L from the area of the deconvolution peaks from Amide I peak. **c**, SEM images of RSF film cryo-fractured cross-sections. The arrow bars in the upper and lower images are 1 μ m and 3 μ m, respectively. **d**, Relative mass change and RH as a function of time measured by DVS.

Changes made to the Supplementary figures:

Fig. S2 | Estimated amorphous and crystalline fractions with changing FWHM. The lower limit of FWHM is fixed to 10/cm, and the upper limits varies for each column: **a, e, i, m)** 20/cm, **b, f, j, n)** 25/cm, **c, g, k, o)** 30/cm (value used for analysis) and **d, h, l, p)** 35/cm. The lower row of graphs, **m, n, o, p)** show proportional areas of the crystalline (β -sheet – red shaded peaks) and amorphous (RC, α -helix and β -turn – blue shaded peaks) contents of Amide I peak deconvolution.

Fig. S3 | Relative mass change and RH as a function of time measured by DVS for all three RSF samples.

4) *In a similar fashion, should Fig. 2d be included in the supplementary information to demonstrate consistency with previous works, or has it already been published elsewhere?*

While Fig. 2d shows alignment with the previous work (Park *et al.*, *Macromol. Rapid Commun.* (2020) - <https://doi.org/10.1002/marc.201900612>), the water sorption isotherms of the three RSF samples we used in this work are not exactly the same as those of previous work, due to the high sensitivity of RSF's secondary structures to the casting and post-treatment conditions. We also believe that it is important to emphasize the low hysteresis of all three RSF samples. For this reason, we have kept Fig. 2d in the main text, and moved Fig. 2c (now Fig. S3) to the Supplementary Information. Changes to both figures are included above.

5) *On page 10, the characterization of H-bonding networks by FTIR is discussed. The terms 'high H-bond density' and 'lower H-bond density' water are used to refer to tightly bound water and bulk-like mobile water respectively. However, it is unclear to me whether these terms can be used interchangeably with strong and weak H-bonds. Indeed, the density of H-bonds in water is not necessarily related to the density of water-biomolecule H-bonds, as shown in Figure 3 of [<https://doi.org/10.1016/j.molliq.2018.10.074>]. Therefore, it is recommended that the use of these expressions be either removed or discussed in greater detail with supporting evidence.*

We thank the reviewer for bringing up this issue. We have removed the “high H-bond density” and “lower H-bond density” in the main text.

Changes made to the main text in the “H-bonding network characterization by FTIR” section:

“To quantitatively understand the properties and the H-bonding networks of water confined in these three RSF samples under different RH conditions, we performed transmission FTIR experiments, which allowed us to estimate the fraction of tightly bound water and bulk-like mobile water from the O-H stretching band (Fig. 4).”

6) *In the same section, the O-H stretching band in FTIR is divided into five populations of H-bonds. However, is this really necessary? The objective is to estimate only the amount of bound and mobile water, which means only two populations are required.*

For example, only two populations (called 'ice-like' and 'liquid-like') are used in the reference cited in the supplementary information, Ref.[16], for increasing RH, although an intermediate population ('transitional') is discussed for intermediate RH. This deconvolution is consistent with the above discussion of bound, unbound, and bulk-like water.

In another example (water confined in micelles) a three-component analysis gives an excellent fit to the data with a minimum number of parameters [<https://doi.org/10.1063/1.1894929>]. Again, this approach is consistent with the bound, unbound, and bulk-like water populations near biomolecules, described above. A deconvolution with three (or possibly two) populations would be sufficient to estimate the B/M water ratio. Additionally, it would be consistent with the ab initio molecular dynamics analysis we conducted for bulk water (Fig. 11 in [<https://doi.org/10.1016/j.molliq.2022.119936>]). In any case, it should be mentioned that the deconvolution with Gaussians is not unique [<https://doi.org/10.1063/1.1894929>].

Moreover, it appears that the five populations' deconvolution is not entirely consistent with the method used in Ref.[38]. In the reference, the fifth population is defined as free-OH, whereas here it is defined as 'water-solid' and associated with a different peak value. To maintain consistency, there should be six populations, including free-OH. It is important to note that the free-OH population is replaced by five-coordinated water when detailed HB-network analysis is performed on bulk water (as shown by the orange triangles in Fig.4 of [<https://doi.org/10.1021/acsnano.0c02984>]).

Furthermore, in Ref. [43], only two populations are used to support the 'water-solid' choice: water-air` interface for strong hydrogen bonds near the hydrophobic interface and fiber-water interface for weaker hydrogen bonds near the hydrophilic interface.

Therefore, how robust is the interpretation of the results presented here if only three populations were used to deconvolute the O-H stretching band (Fig. 4, S3-S5)?

We thank the reviewer for raising the question regarding whether dividing the O-H stretching band in FTIR into five populations of H-bonds is necessary. The selection of the five H-bond populations was determined by establishing a reasonable physical model to approximate the H-bond populations in the samples and ensuring a good fit. We have included a clearer description of our justification with a comparison between alternate protocols for deconvolutions.

To determine the H-bond populations, we first performed the second derivative of the FTIR O-H stretching spectra obtained at both 10% RH and 90% RH (see Fig. S8a), and then compared these second derivatives with the peaks reported previously in literature (Fig. S8b). It is clear that all our silk samples have three major H-bond populations at FTIR peaks of 3210, 3290 and 3410 /cm, which align well with H-bonds found in other materials, including those in ref. 38 (Sun, *Chem. Phys. Lett.* (2013) – <https://doi.org/10.1016/j.cplett.2013.03.065>) (Fig. S8b). Since our RSF samples still contain significant amounts of tightly bound and bulk-like mobile water, evident at both ends of the FTIR spectra, we then assigned two more peaks to represent these less well-defined edge populations. To represent tightly bound water, the 3000/cm peak was chosen as the second derivative characteristic of most RSF samples, representing the low wavenumber population. For bulk-like mobile water, the 3550/cm peak was identified by averaging the high-wavenumber peaks found with the second-order derivative analysis of the three samples at 10% and 90% RH (Fig. S8b). The 3550/cm peak also fairly close to the peaks reported by Ichii *et al.*, Mallamace *et al.* and Laurson *et al.* (3540/cm) and Bernardina *et al.*(3569/cm) (Fig. S8b).

As suggested by the reviewer, we also deconvoluted the O-H stretching band using both three and six populations, as follows.

Three-population deconvolution:

While using the three water populations (3295, 3460 and 3590/cm) suggested in the literature (Brubach *et al.*, *J. Chem. Phys.* (2005) - <https://doi.org/10.1063/1.1894929>), we observed that the deconvolution did not adequately fit the FTIR data (average R^2 0.98768 +/-0.01013 for Silk-H, 0.98234 +/- 0.00969 for Silk-M and 0.942885 +/- 0.00897 for Silk-L), particularly at lower

wavenumbers (see Fig. S9). Whereas the five-population deconvolution yields a much better fit, with average R^2 values of 0.99928 for Silk-H, 0.99935 for Silk-M and 0.99735 for Silk-L. It is most likely because our RSF samples contain a significant amount of bound water, which increases the spectral intensity in the ~2800-3100/cm region. The three-population deconvolution seems to be adequate only for Silk-H at 90% RH, but not for other RSF samples with higher levels of bound water.

Six-population deconvolution:

As suggested by the reviewer, we have also deconvoluted the O-H stretching band using six-populations, with the addition free-O-H stretching peak of 3636/cm found in bulk water (Fig. S10 & Fig. S11). The addition of the 3636/cm does not contribute to the analysis, except for a 0.85% contribution to the total OH-peak area in the case of Silk-L at 10% RH (Fig. S10b). The negligible contribution of the free-OH stretching band to the deconvolution suggests that RSF samples contain a minimal amount of free-OH bonds.

In summary, we believe that the five-population deconvolution is the most reasonable approximation for our RSF samples, based on a combination of second derivative analysis of the FTIR spectra and the overlap of the peak positions found in literature. The three-population deconvolution underperformed in terms of fit quality, while the six-population deconvolution performed similarly to the five-population deconvolution, but the additional free-OH stretching peak did not yield any measurable contribution.

To clarify the protocol that we employed, we have included additional discussions, as well as both the three-population and six-population deconvolutions, in the main text and Supplementary Information.

Changes made to the Methods:

“Five peak positions (AAD - 3000/cm, AADD - 3210/cm, water bound to organic matter - 3290/cm, AD - 3410/cm and ADD - 3550/cm), corresponding five different H-bonding scenarios^{43,44,46-48}, were determined by establishing a reasonable physical model to approximate the H-bond populations in the samples and ensuring a good fit (see Supplemental Note 2 and Fig. S8).”

Changes made to the Supplementary Notes:

“2. Deconvolution of FTIR O-H stretching spectra

To determine the H-bond populations, we first performed the second derivative of the FTIR O-H stretching spectra obtained at both 10% RH and 90% RH (see Fig. S8a), and then compared these second derivatives with the peaks reported previously in literature (Fig. S8b)^{6,7,15,17,18,25,26,28-30}. It is clear that all our silk samples have three major H-bond populations at FTIR peaks of 3210, 3290 and 3410 /cm, which align well with H-bonds found in other materials (Sun, (2013)⁷) (Fig. S8b). Since our RSF samples still contain significant amounts of tightly bound and bulk-like mobile water, evident at both ends of the FTIR spectra, we then assigned two more peaks to represent these less well-defined edge populations. To represent tightly bound water, the 3000/cm peak was chosen as the second derivative characteristic of most RSF samples, representing the low wavenumber population. For bulk-like mobile water, the 3550/cm

peak was identified by averaging the high-wavenumber peaks found with the second-order derivative analysis of the three samples at 10% and 90% RH (Fig. S8b). The 3550/cm peak also fairly close to the previously reported peaks of 3540/cm (Ichii *et al.*²⁶, Mallamace *et al.*¹⁵ and Laurson *et al.*¹⁸) and 3569/cm (Bernardina *et al.*²⁵) (Fig. S8b).

In addition, we have also deconvoluted the O-H stretching band using both three and six populations, as follows.

Three-population deconvolution:

While using the three water populations (3295, 3460 and 3590/cm) suggested in the literature (Brubach *et al.*¹⁷), we observed that the deconvolution did not adequately fit the FTIR data (average R^2 0.98768 +/-0.01013 for Silk-H, 0.98234 +/-0.00969 for Silk-M and 0.942885 +/-0.00897 for Silk-L), particularly at lower wavenumbers (Fig. S9). Whereas the five-population deconvolution yields a much better fit, with average R^2 values of 0.99928 for Silk-H, 0.99935 for Silk-M and 0.99735 for Silk-L. It is most likely because our RSF samples contain a significant amount of bound water, which increases the spectral intensity in the ~2800-3100/cm region. The three-population deconvolution seems to be adequate only for Silk-H at 90% RH, but not for other RSF samples with higher levels of bound water.

Six-population deconvolution:

We have also deconvoluted the O-H stretching band using six-populations, with the addition free-O-H stretching peak of 3636/cm found in bulk water (Fig. S10 & Fig. S11). The addition of the 3636/cm does not contribute to the analysis, except for a 0.85% contribution to the total OH-peak area in the case of Silk-L at 10% RH (Fig. S10b). The negligible contribution of the free-OH stretching band to the deconvolution suggests that RSF samples contain a minimal amount of free-OH bonds.

In summary, the five-population deconvolution is the most reasonable approximation for our RSF samples, based on a combination of second derivative analysis of the FTIR spectra and the overlap of the peak positions found in literature. The three-population deconvolution underperformed in terms of fit quality, while the six-population deconvolution performed similarly to the five-population deconvolution, but the additional free-OH stretching peak did not yield any measurable contribution.

”

Changes made to the Supplementary figures:

Fig. S8 | Deconvolution peak position determination: a) Peak wavenumber and intensity positions of 10 and 90% RH for all three RSF samples obtained through second-order derivative analysis of the OH-stretch peak. b) Comparison of second-order derivative analysis of the sample peaks with the peak positions found in literature^{6,7,15,17,18,25,26,28–30} for bulk and nanoconfined water. The peaks chosen for the deconvolution are 3000, 3210, 3290, 3410 and 3550/cm. The 3000 and the 3550/cm peaks encompass several populations on the edge of the spectra.

Fig. S9 | The deconvolution of FTIR spectra of the Silk-H, Silk-M & Silk-L samples with peak positions from Brubach *et al.*¹⁷ by 3-peak deconvolution: a) at 10% RH. b) at 90% RH.

Fig. S10 | Deconvolution of all three RSF samples at 10% RH with the two peak position protocols. a) 6-peak deconvolution with peak position as found in Sun (2013)⁷ and Igarashi et al. (2020)²¹. **b)** 6-peak deconvolution with peak position from our 5-peak deconvolution with the additional 3636/cm peak.

Fig. S11. | Deconvolution of all three samples at 90% RH with the two peak position protocols: a) 6-peak deconvolution with peak position as found in Sun (2013)⁷ and Igarashi et al.

(2020)²¹. **b)** 6-peak deconvolution with peak position from our 5-peak deconvolution with the additional 3636/cm peak.

7) *The sentence "the amount of the bound water in Silk-H and Silk-M increases with rising RH, while in Silk-L, the amount of the bound water remains relatively constant across the entire RH range from 10% to 90% (Fig. 4c)" appears to contradict Fig. 4d, where the changes in bound water for all three samples are comparable. Is it possible that the authors meant to refer to the structural changes shown in Fig. 3c, which are less noticeable in Silk-L compared to the other two samples?*

This would be more consistent with the subsequent sentence "These differences could be attributed to the increasing H-bonding sites resulting from higher degrees of structural reconfigurations in the amorphous regions of Silk-H and Silk-M during hydration".

However, the authors should also mention that the difference in Fig.3c could be associated with the saturation of the reduced amount of hydrophilic regions exposed to water after the methanol treatment, independent of the RH.

We thank the reviewer for bringing up this point. In Fig. 4c, we intend to show the “absolute” amounts of bound and mobile water contents, while Fig. 4d shows the fractions of bound and mobile water relative to the total water content at 90% RH in each RSF sample. As RH increases from 10% to 90%, the amount of the bound water in Silk-H and Silk-M increases by 81.6% and 51.8%, respectively. In Silk-L the bound water fraction only increases by 11.4%, when compared to the initial bound water content at 10% RH (Fig. 4c). To clarify, we have included quantitative information and revised the discussion in the main text.

As suggested by the reviewer, we have also included a statement that the difference in Fig.3c could be associated with the saturation of the reduced amount of hydrophilic regions exposed to water.

Changes made to the main text in the “H-bonding network characterization by FTIR” section:

“We noted that, as RH increases from 10% to 90%, the amount of the bound water in Silk-H and Silk-M increases by 81.6% and 51.8%, respectively, compared to its initial bound water content at 10% RH, while the amount of the bound water in Silk-L only increases by 11.4% (Fig. 4c).”

Changes made to the main text in the “RSF tryptophan fluorescence and molecular-scale hydration” section:

“Compared to Silk-H and Silk-M, Silk-L have a relatively smaller emission shift, which could be associated with the saturation of the reduced amount of hydrophilic regions exposed to water, after the methanol treatment (Fig. 3c).”

8) *They should also clarify why the methanol treatment should limit the ‘structural reconfiguration in the amorphous regions’? Is it because it favors molten globule structures?*

The exposure of RSF to methanol will lead to interactions with the polar groups of the fibroin macromolecule and thus enhance chain mobility. This enhanced chain mobility allows for the self-assembly of hydrophobic β -sheet nanocrystals, which is a more energy favorable configuration (Wang and Kim *et al.*, PNAS (2020) – <https://doi.org/10.1073/pnas.1911563116> & Rockwood *et al.*, *Nat. Protoc.* (2011) - <https://doi.org/10.1038/nprot.2011.379>). After the methanol treatment, the system is left with much smaller amorphous domains. The reduction of the structural reconfiguration of the amorphous domains is likely due to the reduced domain size together with the constraining of neighboring β -sheet domains.

Changes made to the main text in the “H-bonding network characterization by FTIR” section:

“It also suggests that the fraction of bound and mobile water is variable in Silk-H and Silk-M during hydration and dehydration, resulting from higher degrees of structural reconfigurations in the amorphous regions of Silk-H and Silk-M during hydration. However, the bound water structure remains fixed in Silk-L, likely attributed to the reduced domain size and the constraining of neighboring β -sheet domains, which limit the structural reconfigurations of amorphous domains.”

9) *Furthermore, it should be noted in the manuscript that the fact "that the mobile water dominates the water adsorption and desorption during hydration and dehydration" is consistent with the molecular dynamics calculations performed in [<https://doi.org/10.1016/j.molliq.2018.10.074>] which show that the bound water population is the least affected by dehydration (Fig. 2).*

As suggested by the reviewer, we have made the following changes to the main text:

“The normalized changes in bound and mobile water clearly show that the mobile water dominates the water adsorption and desorption during hydration and dehydration, which is consistent with simulations in previous studies²⁴.”

Changes made to the References and Notes:

“24. Calero, C. & Franzese, G. Membranes with different hydration levels: The interface between bound and unbound hydration water. *Journal of Molecular Liquids* 273, 488–496 (2019).”

10) *The statement "We localize the water exchange predominantly in the hydrophilic amorphous region" and its reiteration in "the B/M ratio of water confined in the amorphous region" cannot be solely based on the present results. As discussed earlier, the collapse in Fig.3d only indicates that water adsorption in the hydrophilic unstructured regions is proportional to the total water content, which is expected. There is no concrete evidence that water predominantly adsorbs in the disordered regions. In principle, water can adsorb in any hydrophilic region, irrespective of its structure. Hence, these sentences need to be rephrased.*

We have rephrased the main text, to align our conclusions with the results we present:

“The energy conversion from the chemical potential of water resulting in mechanical deformation likely occurs within non-crystalline hydrophilic regions, and the high mechanical stiffness resulting from increased crystallinity contributes to the effective transfer of the converted mechanical energy in the amorphous region to external loads.”

“During dehydration, we observe that energy transfer from the chemical potential to macroscopic scale mechanical energy initiates when RSF samples undergo viscous-rubbery phase transitions governed by the B/M ratio of water confined in the material.”

11) According to their observations, the film WR stress possibly saturates when the B/M water ratio exceeds 1.4-1.6, as shown in Figure 1i. They interpret this as being likely due to the maximal mobile water capacity of the unstructured domains. However, this point requires further explanation. It is worth noting that these high values of B/M water ratio are only accessible for the Silk-L sample at the lowest RH, where most water is bound. Under these conditions, any further decrease in RH would not decrease the bound water, which is consistent with [<https://doi.org/10.1016/j.molliq.2018.10.074>]. Therefore, the mechanical response would remain unchanged.

We have revised the explanations in the main text:

“However, we also noted that when the B/M water ratio in Silk-L is higher than 1.4-1.6 at low RH, the WR pressure plateaus. This could be due to the limits of RSF’s structure in translating pressure to external loads.”

12) Minor points:

a) On page 7, in the section titled "Dehydration-induced strain and stress of RSF", there is a typo in the last two lines where Silk-H and Silk-L are interchanged. The same error is also present in the fifth and fourth lines before the last, at the end of the same section on page 8.

Thank you for spotting this error. We have addressed these typos accordingly.

b) DVS stands for dynamic vapor sorption, not dynamic water sorption, as the manuscript uses.

Thank you for pointing out this error. We have corrected the main text, replacing ‘dynamic water sorption’ with ‘dynamic vapor sorption.’

c) The protein regions described as amorphous (unstructured) in this manuscript are commonly referred to as intrinsically disordered regions in protein literature [e.g., [https://doi.org/10.1002/1097-0134\(20010101\)42:1<38::AID-PROT50<3.0.CO;2-3](https://doi.org/10.1002/1097-0134(20010101)42:1<38::AID-PROT50<3.0.CO;2-3)]. Clarifying their synonyms would help connect them with other subfields in protein science.

As suggested by the reviewer, we have made the following additions to the main text:

“For example, within the silk fibroin backbone, the secondary structure of the heavy chain gives rise to both soft, hydrophilic amorphous (intrinsically disordered) regions and stiff, hydrophobic β -sheets nanocrystalline regions^{35,36}.”

d) In section "Viscoelasticity analysis of RSF films" the symbol 'd' in Eq.(2) is not defined. The Eq.(2) should be on a separate line.

We have corrected the typo. The equation is in a separate line and now reads “ $\sigma = E_U \epsilon$.”

e) Due to the reasons explained above, they should avoid referring to their convolutions of "FTIR of the OH stretching band, as it is a well-established method for studying H-bonding of water in materials." Instead, they should cite the references they follow with the above-mentioned caveats.

As suggested by the reviewer, we have deleted the claim and cited the references.

f) The choices of peak values on page 18 for AAD, ADD, and AD do not coincide with those reported in the supporting Ref.[38].

Thank you for pointing this out. The peak values were selected based on a combination of the second derivative analysis of the FTIR spectra and the overlap of the peak positions found in literature. Additional discussions in the main text and the Supplementary Information were included above.

g) In Supplementary Notes, section 1, 'analysis as was' should be 'analysis was'.

We have corrected the error, thank you for pointing it out.

To sum up, the outcome presented in Fig. 1i is quite intriguing as it provides us with a better understanding of the mechanisms regulating the WR of these films. This finding could be relevant in the field of new high-performance actuators. However, the authors are advised to follow the guidelines mentioned above to make the message of this manuscript more robust and clear.

--

*Dr. Giancarlo Franzese, Profesor Titular de Universidad en Física
de la Materia Condensada*

on leave @ Max Planck Inst. for Physics of Complex Systems-Dresden

*Statistical Physics of Bio-Nano Complex Matter Group's P.I.,
Interdisciplinary and Statistical Physics Section--Department of
Condensed Matter Physics (Office 312), Physics &
Institute of Nanoscience and Nanotechnology (IN2UB),*

Universitat de Barcelona, Marti i Franques 1, 08028 Barcelona, Spain
PHONE: +34 93 40 39212 / Secretary: +34 93 40 21150

Chief Editor of the Specialty Section Computational Nanotechnology
in Frontiers of Nanotechnology (open for submission at
<https://www.frontiersin.org/journals/nanotechnology#>)

E-MAIL: gfranzese@ub.edu

ResearcherID: A-9655-2009 Orcid ID: 0000-0003-3006-2766

Scopus Author ID: 7003330838 Loop profile: 109112

BOOK: Aspects of Physical Biology www.springer.com/978-3-540-78764-8

ISSUE: Nonequilibrium Phenomena in Confined Systems mdpi.com/si/7262

WEB: <https://spcmub.wordpress.com/> (with PDF of publications)

Reviewer #2 (Remarks to the Author):

In this work, the authors prepared three types of regenerated silk fibroin proteins to investigate how nanoscale water limitation controls macroscale dehydration-induced actuation in nonporous/nanoporous RSF. It indicates that the increased crystallinities and stiffness will enhance WR pressure and energy density of RSF. However, there are still some issues that have not been well resolved, so I do not recommend the publication of this work. The following issues need to be addressed:

We thank the reviewer for the comments and suggestions that helped to further strengthen the manuscript. We have addressed the reviewer's comments below.

1. In figure 3b, the authors suggest that all RSF samples exhibit a dramatic shift in their tryptophan emission wavelength when RH changes between 10% and 90%. However, neither Silk-M nor Silk-L shows obvious changes and differences, please explain this reason in detail.

Thank you for bringing up this point. As RH increases from 10% to 90%, the emission maxima of Silk-L, Silk-M, and Silk-H increase by 5.5 nm, 12 nm and 16.5 nm, respectively. These changes of Silk-M and Silk-L may be less apparent in the overall fluorescence emission/excitation map in Fig. 3b and Fig. S4. To avoid confusion, we have deleted “dramatic” and revised the discussion in the main text.

Changes made to the main text in the “RSF tryptophan fluorescence and molecular-scale hydration” section:

“When RH changes between 10% and 90%, all RSF samples exhibit a shift in their tryptophan emission wavelength (**Fig. S4**).”

2. In figure 3d, the authors suggest that adsorption/desorption is mainly located in the amorphous regions of RSF samples. Please explain this reason in detail.

Our suggestion that adsorption/desorption mainly occurs in the amorphous regions of RSF samples is evidenced by the following two reasons. First, the shift in the tryptophan emission strongly correlated with the amount of water adsorption across all RSF samples, despite the differences in secondary structures and water adsorption characteristics. This suggests a close proximity between the adsorbed water molecules and tryptophan residues in RSF. Second, since tryptophan residues are mainly localized in the amorphous (linker) regions (Zhou *et al.*, *Proteins* (2001) - doi.org/10.1002/prot.1078), it is likely that the adsorbed water molecules are located in hydrophilic amorphous regions. If the adsorbed water is also located in crystalline regions, the tryptophan emission would show less shift per water molecule uptake in RSF with higher degree of crystallinity. As suggested by the reviewer, we have included these details in the main text.

Changes made to the main text in the “RSF tryptophan fluorescence and molecular-scale hydration“ section:

“Considering that tryptophan residues are exclusively located in the amorphous regions, this observed correlation between the emission shift and water adsorption across all RSF samples suggests that the adsorbed/desorbed water is mainly located in the amorphous regions⁴⁴. Otherwise, the tryptophan emission would show less shift per water molecule uptake in RSF with a higher degree of crystallinity.”

3. In figure 2a, the authors prepared two other films by treating Silk-H films with water and methanol, and realized the modification of the secondary structure of silk fibroin protein. Such operation seems very simple, please explain this mechanism from the molecular level.

Water and methanol treatments are widely used to induce the β -sheet formation in RSF (Wang and Kim *et al.*, PNAS (2020) – <https://doi.org/10.1073/pnas.1911563116> & Rockwood *et al.*, *Nat. Protoc.* (2011) - <https://doi.org/10.1038/nprot.2011.379>). At the molecular level, the exposure of RSF to water and methanol will lead to interactions with the polar groups of the fibroin macromolecule and thus enhance chain mobility. This enhanced chain mobility drives the self-assembly of hydrophobic β -sheet nanocrystals, the more energetically favorable configuration.

To clarify, we have made the following changes to the method section:

“Both water and methanol treatments enhance chain mobility and allow for the self-assembly of hydrophobic β -sheet nanocrystals, which is a more energetically favorable configuration.”

4. In figure 2d, low hysteresis of the measured water sorption isotherms suggests that all RSF films share similar nonporous/nanoporous solid structures. Please add a detailed discussion and characterization of this conclusion.

The hysteresis of water sorption isotherms is generally associated with delayed pore condensation, which is not in thermodynamic equilibrium. If the material contains large pores, cavitation during dehydration also contributes to the hysteresis. The obtained water sorption isotherms do not show any visible hysteresis, and they belong to a traditional type III isotherm

(Thommes *et al.*, *Pure Appl. Chem* (2015) - <https://doi.org/10.1515/pac-2014-1117>), indicative of a nonporous material.

To further characterize the porosity of our RSF samples, we have also conducted SEM imaging of the cross-sections of RSF. These images confirm the absence of visible pores, providing further evidence of the nonporous nature of RSF films.

As suggested by the reviewer, we have included additional discussions and characterizations in the main text and the Supplementary Information.

Changes made to the main text in the “Secondary structure-dependent water adsorption of RSF” section:

“The cross-sections scanning electron microscope (SEM) images and atomic force microscopy (AFM) topographies show that all RSF films share similar non-porous solid structures (Fig. 2c and Fig. S12).”

“The obtained water sorption isotherms show low hysteresis, and they belong to a traditional type III isotherms^{39,40}, providing further evidence of the non-porous nature of RSF films.”

Changes made to the Methods:

“**SEM** -To image the cross-section of RSF films, thin film samples were first submerged in liquid nitrogen for 30 seconds and cryo-fractured to expose their cross-sections. The samples were then coated with ~3 nm of platinum using a sputter coater (EM ACE600, Leica) and imaged using an Environmental SEM (Quattro S, Thermo Scientific).”

Changes made to the references:

“40. Thommes, M. *et al.* Physisorption of gases, with special reference to the evaluation of surface area and pore size distribution (IUPAC Technical Report). *Pure and Applied Chemistry* **87**, 1051–1069 (2015)”

5. This work lacks the application work of materials, and should be supplemented with reasonable and novel application displays, so as to more intuitively and strongly confirm the mechanism study of RSF microstructure in this paper.

We appreciate the reviewer’s suggestion regarding potential practical demonstrations of our material applications. While we recognize the value of application displays, Our focus is the elucidation of the fundamental molecular-scale mechanisms of water responsiveness that help describe and predict the dehydration-induced pressure of silk. To connect this molecular-scale mechanism to a macroscale demonstration, we have prepared a novel application display of the water-responsive actuation of the RSF/polyimide bilayers as a function of RH, which more intuitively illustrates the difference in the actuation capacity of RSF samples, depending on their bound-to-mobile water ratios (Fig. S13). We have included this additional demonstration in the Methods and Supplementary Information.

Changes made to the main text in the Methods section:

“RSF/polyimide bilayers

We demonstrated the RSF samples’ propensity to actuate at different RH levels by preparing RSF/polyimide bilayers. A 6 μm thick layer of RSF was deposited on 12 μm films of plasma-treated (30s in 75% Ar:25% O₂ plasma) polyimide and left to dry overnight. Subsequent samples were then exposed to their respective post-treatments. The sample size was 3 x 6 mm, with a weight of 10.6 mg attached to the end of the bilayer films.”

Changes made to the Supplementary figure:

Fig. S13 | WR actuation of RSF/polyimide bilayers. RSF/polyimide bilayers started to actuate at different RH levels during dehydration. The scale bar is 5 mm.

6. In Figure 2, ATR-FTIR spectra were applied to characterize the secondary structure of the RSF sample before and after water absorption. The characterization of secondary structures is essential for mechanism interpretation, but the structural characterization in this paper is not

persuasive and obvious enough. The evolution of secondary structure could be illustrated by more visual structural characterization methods such as SEM, AFM or SAXS/ WAXS.

We agree that visual structural characterization would be appealing. As suggested by the reviewer, we have performed additional SEM and AFM to characterize the cross-section and surface of RSF film. However, the visual structural characterization of our dense solid-phase material using electron microscopy or force microscopy lacks the resolution to resolve secondary structure. We have included additional discussions and SEM/AFM images in the main text and Supplementary Information. We also agree that the detailed characterization of RSF's secondary structures is an interesting direction for future study. We have applied for Beamline time at the Brookhaven National Laboratory to perform SAXS/WAXS measurements this fall for a separate study on RSF, which is beyond the scope of the current studies.

FTIR and its Amide band deconvolution, which we used in this work, are frequently used to quantitatively characterize the secondary structure of RSF (Belton *et al.*, *Acta Biomaterialia* (2018) - <https://doi.org/10.1016/j.actbio.2018.03.058>). Additionally, the quantitative analysis of secondary structures itself is not essential for our mechanistic interpretation. It shows that, despite the structure differences among these RSF samples, the similar bound-to-mobile water ratio at which all RSF starts to exert force, highlight the dominant role of water structure in water-responsive actuation rather than RSF's secondary structures.

Changes made to the Methods section:

“**SEM** - To image the cross-section of RSF films, thin film samples were first submerged in liquid nitrogen for 30 seconds and cryo-fractured to expose their cross-sections. The samples were then coated with ~3 nm of platinum using a sputter coater (EM ACE600, Leica) and imaged using an Environmental SEM (Quattro S, Thermo Scientific).”

“**AFM** - RSF's surface topographies were imaged by an AFM (Multimode 8, Bruker). RSF films were glued to silicon substrates with double-sided tape. The topographies were imaged by using an AFM probe with a tip radius of ~2 nm (SCANASYST-Fluid, Bruker) in tapping mode.”

Changes made to the main text in the results section, Fig. 2:

Fig. 2 | Secondary structure, SEM and water sorption isotherms of the RSF samples. **a**, Normalized ATR-FTIR spectra of Amide I (1650/cm peak) and II (1520/cm peak) bands and the deconvolution peaks for Silk-H, Silk-M, and Silk-L samples. The deconvolution peaks are centered at 1510/cm (β -sheet), 1530/cm (Random coil), 1540/cm (α -helix), 1565/cm (β -turn), for Amide II peak, 1592/cm (baseline correction – see methods section) and 1620/cm (β -sheet – *inter/intra*-molecular), 1640/cm (α -helix), 1655/cm (Random Coil), 1680 (β -turn), 1695/cm (β -sheet – *inter*-molecular) for the Amide I peak. **b**, Crystalline and amorphous fractions of Silk-H, Silk-M, and Silk-L from the area of the deconvolution peaks from Amide I peak. **c**, SEM images of RSF film cryo-fractured cross-sections. The arrow bars in the upper and lower images are 1 μ m and 3 μ m, respectively. **d**, Relative mass change and RH as a function of time measured by DVS.

Changes made to the Supplementary figure:

Fig. S12. | AFM topographies of the RSF samples surface. Scale bar is 1 μ m.

7. *This work investigates the macroscale dehydration-induced actuation in nonporous/nanoporous RSF and the effect of B/M ratio on structure. There are many similar studies. Please explain the fundamental innovation that distinguishes this study from other studies.*

We appreciate the reviewer's comments. Our novel fundamental finding is the molecular-scale connection between bound/mobile water populations and water-responsive pressure. We are not aware of any study on RSF's or other water-responsive material's that focus on the quantitative understanding of bound and mobile water populations and their effects on these materials' water-responsive pressure. We are aware of seminal studies identifying different water populations in RSF via NMR techniques (Asakura *et al.*, *Acta Biomaterialia* (2017) - <https://doi.org/10.1016/j.actbio.2016.12.052>); however, not in the context of water-responsive properties.

The fundamental innovation of our study is the elucidation of the crucial role of bound-to-mobile water in determining the macroscopic water-responsive properties of RSF, regardless of its secondary structures. Coupled with a straightforward standard linear solid model, our findings provide a strategic approach to designing materials' water-responsiveness, which was not possible before.

Reviewer #3 (Remarks to the Author):

This manuscript attempts to illustrate the underlying mechanisms of deformation in response to relative humidity of biological water-responsive materials. They found that the nanoscale confinement of water dominates the macroscopic dehydration-induced stress of the regenerated silk fibroin. Silk samples start to exert force as they undergo phase transitions from a viscous to rubbery state. These transitions occur when the bound-to-mobile (B/M) ratio of confined water reaches the same level. This critical B/M water ratio suggests a common threshold above which the chemical potential of water instigates the work of actuation in silk. Their findings may be beneficial to develop new physics to describe the WR behavior of biopolymers through confined water.

They've done a very professional work, and reasonably illustrated the underlying mechanisms of deformation in response to relative humidity of biological water-responsive materials. However, I don't think their achievement is enough to be published in Nat. Commun.

We thank the reviewer for acknowledging the potential benefits of developing new physics and illustrating the underlying mechanisms of biological materials' water-responsive behavior. We also thank the reviewer for raising concerns regarding the impact of our work. We are grateful for this opportunity to better explain the novelty and explain the far-reaching potential of the fundamental new insights.

1. *This work revolves around the underlying mechanisms of deformation in response to relative humidity of biological water-responsive materials. A great deal of characterization and analytical work was used to reveal the mechanism. However, in my opinion, water-responsive*

actuation of silk is not much different from other common water-responsive materials such as hydrogels and elastomers. The driving force of deformation mainly attribute to the uneven swelling from uneven distribution of water. Therefore, I can't find any important advance or novel theory in the manuscript.

The macroscale deformation of the material in our system (as well as hydrogel/elastomer systems) can indeed be attributed to the uneven distribution of swelling. In contrast, our investigation focuses on the molecular-scale origin of 'extreme' water responsive energy densities, particularly for biomolecular systems. The work reported here reveals new fundamental understanding of bound-to-mobile water in determining the macroscopic water-responsive properties of RSF, regardless of its secondary structures or water distribution. Additionally, when this molecular-scale perspective is coupled to a standard linear solid model, our findings provide a strategic approach to predicting materials' hydration-induced stress, which was not possible before.

We appreciate the comment, and we have reiterated the novelty of our new mechanistic understanding of a materials' water-responsiveness, as follows:

a) To our knowledge, there is no universal physics or theory capable of explaining or predicting the stress generated during the water-responsiveness of materials, especially for nonporous soft matter exhibiting significant water-responsive strain. As highlighted in recent review articles (Gor, Huber, and Bernstein, *Appl. Phys. Rev.* (2017) - <https://doi.org/10.1063/1.4975001>, Quan *et al.*, *Nature Reviews Materials* (2021) - <https://doi.org/10.1038/s41578-020-00251-2>), some thermodynamic theories can quantitatively predict adsorption-induced stresses and strains of porous materials, which exhibit relatively low strains associated with the deformation. However, for biomaterials and other soft matter with large water-responsive strain (>10%), existing models are inadequate and unable to predict their water-responsive behavior, including the stress generated during hydration processes.

b) The findings presented in this work can make a significant contribution to advancing new physics and developing more precise models aimed at understanding water-responsiveness, particularly through consideration of confined water properties. Recent studies on high-energy density water-responsive materials (Piotrowska *et al.*, *Nature Materials* (2021) - <https://doi.org/10.1038/s41563-020-0799-0>, Harrellson *et al.*, *Nature* (2023) - <https://doi.org/10.1038/s41586-023-06144-y>, Wang and Liu *et al.*, *Advanced Science*, (2022) - <https://doi.org/10.1002/advs.202104697>) have shown the unconventional properties of water confined within these materials and the importance of these properties in defining materials' water-responsive behaviors. This paper advances the field by proposing a novel approach that quantitatively explains the dehydration-induced pressure through connections with water structure. More importantly, as recognized by the first reviewer, "the existence of a critical bound-to-mobile water value that determines the macroscopic properties of films, regardless of the proteins' microscopic structure, offers a strategy for material design in high-performance actuators."

c) The uneven swelling resulting from an uneven distribution of water can lead to different actuation modes, such as bending, curling, and twisting (Park and Chen, *J. Mater. Chem. A*

(2020) - <https://doi.org/10.1039/D0TA02896G>). However, it cannot explain or predict the water-responsive behavior of the material itself. The uneven swelling is also not a critical requirement for these hygroscopic materials, including hydrogels and elastomers, to exert force during water-responsiveness.

We therefore believe that our work adds an important and novel element to understanding and predicting the water-responsive behaviors of hygroscopic materials. To highlight the novelty, we have included additional discussions in the main text.

Changes made to the Summary:

“However, the underlying mechanisms of their forceful actuation are not fully understood, and there is a lack of physics or theories capable of explaining or predicting the stress generated during water-responsiveness.”

Changes made to the main text in the Introduction:

“Additionally, there is a lack of reliable methods for predicting their swelling/shrinking strain and stress when subjected to RH changes, which impedes the use and engineering of WR materials².”

“While these attempts can describe swelling behaviors for some porous materials with relatively low WR strains^{7,17}, it remains difficult to predict WR behavior of biomaterials and other non-porous soft matter, particularly those with strong material-water interactions and bonding reorganization that lead to a dramatic change in the chemical potential of confined liquids during WR actuation^{2,17}.”

Changes made to the main text in the Discussion:

“Our study shows a quantitative approach explaining macroscale dehydration-induced pressure in non-porous RSF through connections with water structure.”

2. The authors say their findings can act as guidelines for predicting and engineering silk's WR behavior, however, it's not proven that how can effectively regulate silk's WR behavior mode based on their findings.

The water-responsive behavior examined in this work pertains to dehydration-induced stress and energy conversion, rather than the water-responsive actuation modes such as bending, curling, and twisting, which can be achieved through uneven swelling or the design of mechanical geometries. We have revised the manuscript to clarify this point.

Changes made to the main text in the Discussion section:

“Our findings provide a strategic approach to controlling and predicting the stress generated during hydration and dehydration of biopolymers for diverse engineering applications.”

In a word, I don't deny that this is meaningful work, but it is not enough published in this journal.

We hope that our explanations effectively communicate our points to the reviewer.

Reviewer #4 (Remarks to the Author):

We would like to thank the reviewer for their contributions to the review process.

REVIEWER COMMENTS

Reviewer #1 (Remarks to the Author):

The authors took into consideration my comments and modified the manuscript in accordance. All the answers to my questions and those of the other referees clarified my doubts, except for the one regarding point 7 in my report. Specifically, after their reply, I understand Fig. 4c, but now I am confused about Fig. 4d. They mention that "Fig. 4d shows the fractions of bound and mobile water relative to the total water content at 90% RH." However, this statement seems contradictory to the caption in the manuscript, which reads: "d, The relative content changes of total, mobile, and bound water normalized to the total water content of each RSF sample at 90% RH." Is Fig. 4d showing the 'fractions' or the 'relative content changes' of bound and mobile water relative to the total water content at 90% RH?

For better clarity, I recommend they include the formula used to define the quantity in Fig. 4d.

With the modifications suggested and the authors' responses, I am confident that the manuscript is now ready for publication. The reasons for this readiness have been detailed in my previous report, and I believe that the manuscript is now in a strong position for publication.

Reviewer #2 (Remarks to the Author):

This work has conducted detailed research in the field of material response and actuation. However, I believe the manuscript lacks the level of innovation required for publication in Nature Communications. I suggest the authors consider submitting it to another journal that is more suitable for this topic.

Reviewer #3 (Remarks to the Author):

I spent a lot of time trying to uncover the revised manuscript's improvement, I don't think a large increase in citations can significantly improve the scientific quality of the article. This work comprehensively demonstrates the relationship of water adsorption rate of the silks and B/M ratio of confined water with the mechanical property of the silks, in this respect it has been very successful. However, I still can't recommend its publication in such a high-impact journal. Because there is no unexpected new theory or great science advances in the manuscript, and the added RSF/polyimide bilayers application demonstration is featureless, all the water absorbent material can exhibit the similar performances. I don't deny the significance of this work, I just don't think it fits the level of this journal in my mind.

Reviewer #4 (Remarks to the Author):

Reviewer #1:

The authors took into consideration my comments and modified the manuscript in accordance. All the answers to my questions and those of the other referees clarified my doubts, except for the one regarding point 7 in my report. Specifically, after their reply, I understand Fig. 4c, but now I am confused about Fig. 4d. They mention that "Fig. 4d shows the fractions of bound and mobile water relative to the total water content at 90% RH." However, this statement seems contradictory to the caption in the manuscript, which reads: "d, The relative content changes of total, mobile, and bound water normalized to the total water content of each RSF sample at 90% RH." Is Fig. 4d showing the 'fractions' or the 'relative content changes' of bound and mobile water relative to the total water content at 90% RH?

For better clarity, I recommend they include the formula used to define the quantity in Fig. 4d.

With the modifications suggested and the authors' responses, I am confident that the manuscript is now ready for publication. The reasons for this readiness have been detailed in my previous report, and I believe that the manuscript is now in a strong position for publication.

We appreciate the detailed review and valuable feedback from the reviewer. We apologize for the typo. Fig. 4d shows the relative water content changes. To clarify, we have included the formula in the Methods section, as suggested by the reviewer.

The following changes have been made to the Methods section (FTIR spectroscopy):

For Fig.4d, the relative water content changes in bound and mobile water were obtained by using the following formula:

$$\frac{A_X(\text{RH}) - A_{X,i}(10\% \text{ RH})}{A_T(\text{RH}) - A_{T,i}(10\% \text{ RH})} \times 100 (\%) \quad (3)$$

where $A_X(\text{RH})$ represents the amount of bound or mobile water and $A_T(\text{RH})$ represents the total amount of water at a given RH value, with the $A_{X,i}(10\% \text{ RH})$ and $A_{T,i}(10\% \text{ RH})$ indicating their initial values at 10% RH.

Reviewer #2:

This work has conducted detailed research in the field of material response and actuation. However, I believe the manuscript lacks the level of innovation required for publication in *Nature Communications*. I suggest the authors consider submitting it to another journal that is more suitable for this topic.

We appreciate the feedback. However, we disagree with the reviewer's opinion on "the manuscript lacks the level of innovation required for publication in *Nature Communications*." Our study has, for the first time, established a novel, quantitative approach that correlates the confined water structure with macroscopic water-responsive properties of the material. We have shown that water structure is the determining parameter governing dehydration-induced stress, regardless of material structure. This discovery represents an important advancement in predicting and engineering the properties of hygroscopic materials, including bio-materials, hydrogels, and polymers that deform in response to humidity changes. This important new physics and mechanistic insight was also recognized by Reviewer

#1: “The researchers establish an interesting relationship between micro and macro properties, which I believe is significant enough to be published.”

Regarding the specific issues raised by the reviewer previously, we have meticulously addressed all the comments. The additional experiments and analyses that resolve the identified concerns have significantly strengthened the manuscript and ensured that the work meets the publication standards of high-impact journals.

Overall, our findings are relevant not only for the field of hydroscopic materials, but also for broader areas studying stimuli-responsive matter and material-water interactions. The insights gained and the experimental approaches we developed could be applied to fundamental research involving other hydroscopic materials as well as for applications including tissue engineering, biomedical and biocompatible material engineering, optical coatings, food preservation, and cosmetics.

We firmly believe that the scope and impact of our work align well with the objectives of *Nature Communications*, as our research not only advances the fundamental understanding of an important and increasingly studied RSF materials but also has broader practical implications relevant to the journal’s wide readership.

Reviewer #3:

I spent a lot of time trying to uncover the revised manuscript's improvement, I don't think a large increase in citations can significantly improve the scientific quality of the article. This work comprehensively demonstrates the relationship of water adsorption rate of the silks and B/M ratio of confined water with the mechanical property of the silks, in this respect it has been very successful.

We appreciate the time and effort the reviewer has put into reviewing our revised manuscript, as well as the encouraging words. We would like to highlight that besides the additional citations, we have conducted new experiments and quantitative analyses to address the comments from the reviewers. We believe these revisions have significantly strengthened the manuscript and ensured that the work meets the publication standards of high-impact journals.

However, I still can't recommend its publication in such a high-impact journal. Because there is no unexpected new theory or great science advances in the manuscript, and the added RSF/polyimide bilayers application demonstration is featureless, all the water absorbent material can exhibit the similar performances.

We respectfully disagree with the reviewer’s opinion on “there is no unexpected new theory or great science advances in the manuscript.” We would like to highlight the important fundamental new knowledge and theory:

- l) Our study has, for the first time, established a novel, quantitative approach that correlates the confined water structure with the macroscopic water-responsive properties of the material. To our knowledge, no existing physics or theory can explain or predicting the stress generated during water-responsiveness in biomaterials and other soft matter with large water-responsive strains (>10%).

- II) Previous studies over the past decades on silk and other hygroscopic materials have focused on qualitative explanations of hydration- and dehydration-induced stress from the materials' structural point of view. Our studies quantitatively demonstrates that water structure is the determining parameter governing dehydration-induced stress, regardless of the material's structure.
- III) We also demonstrate the crucial role of water structure in the materials' hydration-dependent viscoelastic properties, suggesting a new approach to quantitatively correlate mechanical properties and water-responsive actuation.

While the assessment of “great science” is inherently subjective, we believe the three points above contribute to a “new theory” for understanding and predicting water-responsive actuation.

We'd also like to clarify that the addition of the RSF/polyimide bilayer demonstration addresses the previous comments from Reviewer #2. This demonstration aims to visually and intuitively illustrate how these silk samples begin to exert force at different humidity levels. We show that the evolution of force corresponds to a common bound and mobile water ratio. This work is not meant to showcase the high performance of silk fibroin films. As noted by the reviewer, the fact that all (or many) water-absorbent materials show water-responsive actuation highlights the broad relevance of our fundamental study.

I don't deny the significance of this work, I just don't think it fits the level of this journal in my mind.

We thank the reviewer for acknowledging the significance of our work. However, we respectfully disagree with the reviewer's opinion on “I just don't think it fits the level of this journal in my mind.”

We believe that a study addressing the fundamental understanding of the mechanical and water-responsive properties of a widely studied material, silk fibroin, will have a significant impact in the field. For example, since the submission of this manuscript in January 2024, 13 papers have been published in *Nature Communications* alone on various properties and applications of silk fibroin materials. While our study focuses on silk, the fundamental knowledge and experimental approaches are also valuable for other hygroscopic materials, with applications including tissue engineering, biomedical and biocompatible material engineering, optical coatings, food preservation, and cosmetics.

Given the scientific novelty and broader impact of our work, we firmly believe that it aligns well with the objectives of *Nature Communications* and meets the journal's high standards.

Reviewer #4:

We thank the reviewer for their contribution to the review process.

REVIEWERS' COMMENTS

Reviewer #1 (Remarks to the Author):

The authors have effectively addressed my last concern and made the necessary modifications to the manuscript.

As I wrote in my initial report, the existence of a critical B/M water value that determines the macroscopic properties of films, regardless of the proteins' microscopic structure, offers a strategy for material design in high-performance actuators. This is a noteworthy result. It makes the work significant to its field and related fields, including tissue engineering, biomedical and biocompatible material engineering, optical coatings, food preservation, and cosmetics.

The authors have correctly identified the water structure as the key parameter governing dehydration-induced stress, irrespective of the material's structure. This finding is paramount because it suggests that all (or many) water-absorbent materials can have water-responsive actuation, making the result of broad relevance for other studies.

The role of confined water in biology, biomaterials, and nanotechnology still needs to be valued and addressed. Rigorous experiments like this are required to finally show beyond any reasonable doubt what simulations already support: nano-confined water's central role in determining macroscopic mechanical properties.

For this reason, the manuscript is in a strong position for publication in Nature Communication.